# AI Security in the Foundation Model Era: A Comprehensive Survey from a Unified Perspective

**Zhenyi Wang**                                                        *zhenyi.wang@ucf.edu*
*Department of Computer Science, Institute of Artificial Intelligence*
*University of Central Florida*

**Siyu Luan**                                                              *siyu.luan@sund.ku.dk*
*University of Copenhagen*

**Reviewed on OpenReview:** *https://openreview.net/forum?id=1g7pKgClZs*

## Abstract

As machine learning (ML) systems expand in both scale and functionality, the security landscape has become increasingly complex, with a proliferation of attacks and defenses. However, existing studies largely treat these threats in isolation, lacking a coherent framework to expose their shared principles and interdependencies. This fragmented view hinders systematic understanding and limits the design of comprehensive defenses. Crucially, the two foundational assets of ML—**data** and **models**—are no longer independent; vulnerabilities in one directly compromise the other. The absence of a holistic framework leaves open questions about how these bidirectional risks propagate across the ML pipeline. To address this critical gap, we propose a *unified closed-loop threat taxonomy* that explicitly frames model–data interactions along four directional axes. Our framework offers a principled lens for analyzing and defending foundation models. The resulting four classes of security threats represent distinct but interrelated categories of attacks: (1) Data→Data (D→D): including *data decryption attacks and watermark removal attacks*. (2) Data→Model (D→M): including *poisoning, harmful fine-tuning attacks and jailbreak attacks*; (3) Model→Data (M→D): including *model inversion, membership inference attacks, and training data extraction attacks*; (4) Model→Model (M→M): including *model extraction attacks*. We conduct a systematic review that analyzes the mathematical formulations, attack and defense strategies, and applications across the vision, language, audio, and graph domains. Our unified framework elucidates the underlying connections among these security threats and establishes a foundation for developing scalable, transferable, and cross-modal security strategies, particularly within the landscape of foundation models.

## 1 Introduction

The growth of machine learning (ML) has brought about not only more powerful and versatile systems but also an increasingly intricate security landscape. A wide spectrum of threats has emerged, including poisoning, evasion, extraction, and inference attacks, alongside a variety of defensive strategies designed to counter them. While these contributions have advanced the field, they are often examined in isolation, emphasizing case-specific mechanics rather than uncovering the underlying principles that connect them. This siloed treatment fragments our understanding of adversarial behaviors, complicates efforts to reason about their relationships, and hinders the development of defenses that remain effective across diverse attack surfaces. In practice, both researchers and practitioners are left without a coherent framework to navigate the accelerating expansion of ML vulnerabilities.

Underlying this complexity is the fact that the two essential building blocks of ML—**data** and **models**—are deeply interdependent. Compromising data integrity can destabilize or corrupt models, while weaknesses in

models can expose private data or propagate errors downstream. Yet, existing surveys rarely capture this mutual influence or explain how risks circulate through the end-to-end ML pipeline. This gap is especially pressing in the context of foundation models, which underpin a wide range of applications and amplify the consequences of security breaches. To address this challenge, we introduce a *unified closed-loop threat taxonomy* that characterizes security dynamics along four directional flows between data (D) and model (M): Data → Data (D→D), Data → Model (D→M), Model → Data (M→D), and Model → Model (M→M), as illustrated in Figure 1.

The resulting four classes of security threats represent distinct but interrelated categories of attacks: (1) D→D: This category encompasses attacks that directly manipulate or recover data content. A *data-decryption attack* attempts to recover plaintext information without access to the secret key; a *watermark-removal attack* seeks to eliminate embedded provenance or ownership identifiers; (2) D→M: *poisoning and harmful fine-tuning attacks* injects malicious samples into training pipelines to induce targeted model misbehavior; and a *jailbreak attack* constructs adversarial inputs that bypass safety mechanisms, causing the model to disregard policy constraints and carry out unintended behaviors; (3) M→D: *model inversion, membership inference attacks, and training data extraction attack* reconstructs sensitive training content from the model's outputs/representations or infers data membership in the training data; (4) M→M: *model extraction attack* replicates proprietary models via limited model queries.

These interdependencies are not merely incidental, they form a dynamic chain of influence wherein the compromise of one component (data or model) can recursively propagate vulnerabilities throughout the ML pipeline. Our study offers a comprehensive review of existing research, detailing the mathematical formulations, adversarial and defensive methodologies, and applications spanning visual, linguistic, auditory, and graph-structured data. They jointly form a closed loop of data and model interactions. Although each threat category corresponds to a distinct class of attacks, these attacks are deeply interconnected and often influence one another in non-trivial ways. On the one hand, certain attacks can amplify others. For example, data poisoning attacks (D→M) can significantly increase a model's vulnerability to membership inference (M→D) (Chen et al., 2022; Wen et al., 2024b). Likewise, Carlini et al. (2021) show that training data extraction can be reduced to membership inference, further reinforcing the practical linkage between these threat categories. Similarly, successful model extraction (M→M) not only enables downstream attacks such as training data recovery and model inversion (M→D) (Tramèr et al., 2016), but can also be leveraged to synthesize adversarial samples (D→D) (Papernot et al., 2017). On the other hand, attacks may also interfere with or weaken one another. For instance, (Wang et al., 2025) demonstrate that introducing backdoor attacks (D→M) can reduce the effectiveness of subsequent model extraction (M→M), highlighting that interactions across attack stages are not always additive. Together, these examples illustrate that security threats should not be analyzed in isolation, but rather as components of a tightly coupled and dynamic closed-loop system.

As these threats cascade, treating each attack as independent is inadequate. Instead, they motivate a global framework that captures the *full feedback loop* between data and model. This closed-loop perspective is essential for designing robust, scalable, and cross-modal defenses, particularly in the foundation model era, where data and model boundaries are deeply entangled and increasingly inseparable.

To highlight the unique scope and perspectives of our work that are not addressed in existing literature, we provide a comprehensive comparison with prior surveys in Table 1. Existing surveys typically focus on individual attack categories in isolation, lacking a holistic or closed-loop perspective. As a result, they do not systematically characterize the interdependencies among different security threats or how vulnerabilities propagate across stages of the learning pipeline. In contrast, our survey covers all four categories of data and model interaction attacks, providing a unified and global perspective that reveals how different attack surfaces interconnect and influence one another in foundation models.

Our contributions can be summarized as:

1. We provide a comprehensive survey that unifies all four data–model attack directions, offering a holistic perspective on the interconnections across threat surfaces.

2. We systematically categorize and analyze representative attacks and defenses within a closed-loop framework, highlighting common principles, differences, and emerging patterns.

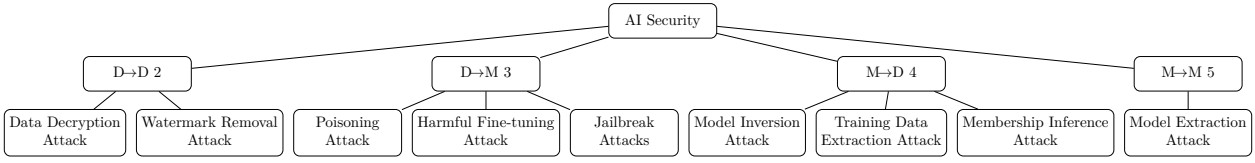

Figure 1: Taxonomy of attacks in AI security

Table 1: **Comparison with prior surveys and taxonomies.** Coverage across closed-loop attack directions and emphasis on foundation-model (FM), closed-loop analysis.

| Survey | D→D | D→M | M→D | M→M | Loop | FM | Scope and limitations |
|---|---|---|---|---|---|---|---|
| **Training data poisoning survey** (Tian et al., 2022) | × | ✓ | × | × | × | × | data poisoning attack |
| **Model inversion attack survey** (Fang et al., 2024) | × | × | ✓ | × | × | ✓ | model inversion attack |
| **Membership inference attack survey** (Hu et al., 2022a) | × | × | ✓ | × | × | ✓ | membership inference attack |
| **Harmful fine-tuning attack** (Huang et al., 2024b) | × | ✓ | × | × | × | ✓ | large language model harmful fine-tuning attack |
| **Jailbreak attack** (Yi et al., 2024b) | × | ✓ | × | × | × | ✓ | large language model jailbreak attack |
| **Model extraction (stealing) survey** (Oliynyk et al., 2023) | × | × | × | ✓ | × | × | Deep coverage of model extraction and Intellectual Property (IP) theft |
| **Our survey (closed-loop taxonomy)** | ✓ | ✓ | ✓ | ✓ | ✓ | ✓ | Unifies threats as data-model feedback loops; explicitly models bidirectional propagation across the full foundation model pipeline and multiple modalities |

3. We identify open challenges and promising research directions, providing guidance for developing more generalizable and resilient defenses against future threats.

**Paper Organization.** Section 2 introduces D→D threats, including data decryption and watermark removal attacks and defenses. Section 3 surveys D→M data poisoning, harmful fine-tuning and jailbreak attacks and defenses. Section 4 explores M→D threats such as model inversion, membership inference, and training data extraction attacks and defenses. Section 5 reviews M→M attacks, encompassing data-free, data-based, and functionality and architecture cloning attacks and defenses. Section 6 presents attacks and defense comparisons, interactions and inter-dependencies. Section 7 discusses other orthogonal views of AI security. Section 8 discusses open challenges and future directions. Section 9 evaluates AI security empirically. Finally, Section 10 concludes the paper by summarizing the unified framework and broader impact.

## 2 Data→Data (D→D)

### 2.1 Protection-Bypass Attacks

Protection-bypass attacks (Zou et al., 2022; 2025) have become increasingly prevalent in the modern machine learning era, as unauthorized access, misuse, or circumvention of protective mechanisms can lead to severe security and ethical risks. These attacks generally aim to transform protected data to bypass ownership

constraints—such as digital encryption and watermarking. Since classical adversarial attacks are already well established and extensively surveyed in the literature (Xiao et al., 2018a;b; Xu et al., 2020; Chakraborty et al., 2021), we do not emphasize them in this survey. Instead, we focus on two representative categories of protection-bypass attacks: *data decryption attacks* and *watermark removal attacks.*

The goal of *data decryption attacks* (Laad & Sawant, 2021) is to transform encrypted datasets in order to gain unauthorized access or use. Such attacks attempt to recover the original data $\boldsymbol{x}$ from its protected representation $\tilde{\boldsymbol{x}}$ without possessing the secret key.

The goal of *watermark removal attacks* (Zhao et al., 2024a) is to transform a watermarked data $\tilde{\boldsymbol{x}}$ (image, or text) into another data $\boldsymbol{x}'$ that preserves task utility while disabling watermark detection or decoding. The attack operates solely through low-level (e.g., pixel- or feature-level) or content-level transformations of the dataset, without accessing the model parameters. The scope includes visible marks (logos, stamps, overlays) and invisible marks embedded in the spatial, frequency, or learned feature domains.

*Common Procedures.* Despite their differences, these attacks share a common operational structure. Attackers typically optimize a transformation that maps a protected input $\tilde{\boldsymbol{x}}$ to an output $\boldsymbol{x}'$, satisfying two conditions: (i) the ownership constraint is removed (e.g., watermark undetectable, cipher broken, or safety bypassed); (ii) the transformed data retains high visual or semantic fidelity to the original. For *image data*, typical pipelines first localize protected regions (e.g., watermarked or encrypted areas) and then restore or regenerate them. Traditional signal-processing operations such as compression, denoising, and filtering remain strong baselines for disrupting watermark detectors, while modern attacks increasingly use generator-based regeneration or latent-space resampling (e.g., encode–decode pipelines with diffusion models or VAEs) to perform end-to-end rewriting (Zhao et al., 2024a). For *text data*, token-level perturbations (insertion, deletion, substitution), paraphrasing, back-translation, or guided rewriting can weaken watermark statistical signals and evade detection (Piet et al., 2025). These rewriting-based transformations leverage semantic-preserving reformulations to bypass ownership constraints while maintaining fluency and meaning.

## 2.2 Mathematical Formalization

General Objective: Let $\boldsymbol{x} \in \mathcal{X}$ denote a clean data sample. A protection mechanism $P$ transforms $\boldsymbol{x}$ into a protected form $\tilde{\boldsymbol{x}}$ by applying ownership or safety constraints such as encryption, watermarking: $\tilde{\boldsymbol{x}} = P(\boldsymbol{x}; \boldsymbol{k})$, where $\boldsymbol{k}$ denotes a secret key (for encryption and authorized decryption), a watermark identifier, or may be null (so $P$ reduces to the identity). The adversary aims to construct an operator $U \in \mathcal{U}$ that maps $\tilde{\boldsymbol{x}}$ to a surrogate $\boldsymbol{x}' = U(\tilde{\boldsymbol{x}})$ that (i) recovers the original content without the decryption key or (ii) removes or invalidates embedded ownership marks. The attack objective can be written as:

$$\max_{U \in \mathcal{U}} \quad O[V(U(\tilde{\boldsymbol{x}})) = 1] \tag{1}$$
$$\text{s.t.} \quad S(\boldsymbol{x}', \boldsymbol{x}) \geq \tau, \qquad C(U) \leq B.$$

Here, $V$ denotes a verification function that outputs 1 when the attack is successful and 0 otherwise. The probability operator $O$ is taken over the data distribution $\boldsymbol{x}$, representing the likelihood that verification still succeeds after the adversarial transformation. Specifically: for *(i) data decryption attacks*, $V$ verifies whether unauthorized decryption succeed. A successful attack means that the original data can be recovered without the secret key $\boldsymbol{k}$. For *(ii) watermark removal attacks*, $V$ verifies whether the embedded watermark remains detectable. A successful attack means that the watermark becomes undetectable.

**Special Case I   Data Decryption Attacks (Laad & Sawant, 2021).** When the protection mechanism is encryption, $U = D$ is an decryption function: $\boldsymbol{x}' = D(\tilde{\boldsymbol{x}})$ and $V$ verifies whether the recovered sample $\boldsymbol{x}'$ successfully reconstructs the original data, implying that the encryption has been effectively reversed. The attack seeks to approximate the decryption process without access to the secret key, producing $\boldsymbol{x}' \approx \boldsymbol{x}$.

**Special Case II   Watermark Removal Attacks (Zhao et al., 2024a).** When the protection mechanism is watermarking, $U = A$ is a watermark removal attacks, i.e., $\boldsymbol{x}' = A(\tilde{\boldsymbol{x}})$ and $V$ is a detection or decoding function $D$ that verifies the presence of watermark. A successful attack means that the embedded watermark is no longer detectable. At the same time, $\boldsymbol{x}'$ should preserve utility for downstream tasks.

### 2.3 Taxonomy and Techniques of Protection-Bypass Attacks

We broadly divide the D→D attacks into two major categories: *1) Data Decryption Attacks*, which attempt to recover original content from encrypted data; and *2) Watermark Removal Attacks*, which aim to erase ownership signals embedded in data.

#### 2.3.1 Data Decryption Attacks

These attacks aim to recover or approximate original data from encrypted data without possessing the secret key. We categorize them into four main families according to strategy and adversarial assumptions: *(a) key-recovery and cryptanalysis*, exploiting brute-force, weak diffusion, or statistical dependencies in chaos-based cryptosystems (Guan et al., 2005; Fridrich, 1998); *(b) ciphertext-only and statistical reconstruction*, where attackers infer data distributions or visual content directly from ciphertext features, such as in GAN- or feature-based ciphertext-only attacks (Sirichotedumrong & Kiya, 2020); *(c) generative-model and learning-based regeneration*, leveraging pretrained generative priors (e.g., GANs or diffusion models) to reconstruct visually plausible plaintexts (MaungMaung & Kiya, 2023); and *(d) side-channel and leakage-based attacks*, where partial computational leakage (e.g., timing or memory access) undermines key secrecy or enables partial recovery (Benhamouda et al., 2018).

#### 2.3.2 Watermark Removal Attacks

Watermark removal seeks to erase ownership signals while maintaining perceptual or semantic fidelity. Approaches can be roughly divided by operating space: *(a) pixel- and signal-space distortions* apply JPEG compression, filtering/denoising, noise injection, resampling, rotation, scaling, cropping, or affine transforms as classical baselines (Wan et al., 2022; Begum & Uddin, 2020; Mousavi et al., 2014); *(b) Mask-guided detection and inpainting* methods adopt a two-stage "localize–then–restore" paradigm based on decomposition or refinement networks. In the first stage, the attacker explicitly detects or localizes the watermark region by generating a mask that estimates the opacity, color, or spatial extent of the mark. In the second stage, the masked area is filled in by an inpainting or restoration network to recover the original image content hidden beneath the watermark (Liu et al., 2021; Liang et al., 2021; Zhao et al., 2022; Niu et al., 2023). *(c) Generator-based regeneration*, where architectures such as convolutional, transformer, or disentangled networks (Li et al., 2021a; Sun et al., 2023) learn a single-branch end-to-end mapping that directly translates a watermarked input into a clean output without producing any intermediate mask, and *(d) latent-space resynthesis*, where diffusion or VAE models remove watermarks in the representation space (Su & Zhang, 2025; Zhao et al., 2024a; Liu et al.), this process effectively bypasses localized pixel- level traces of the watermark. For text, *(e) editing- and paraphrasing-based attacks*, which weaken watermark signals by applying semantics-preserving transformations such as word substitution, paraphrasing, or style rewriting that alter token statistics or sentence structure to evade detection (Yang et al., 2025b; Kirchenbauer et al., 2023; Liu et al., 2024a), while *(f) Model-driven approaches* infer or neutralize watermarking rules directly through surrogate modeling or decoding-time neutralization (Pan et al., 2025). Recent surveys highlight a paradigm shift from heuristic distortions toward learning-based regeneration methods that balance watermark removal effectiveness with perceptual or semantic fidelity (Su & Zhang, 2025; Liu et al., 2024a; Wan et al., 2022).

### 2.4 Defensive Techniques

Defenses against data→data attacks share a common goal: preserve data ownership and safe use under strong, adaptive adversaries. We organize defenses into two families aligned with the attacks reviewed above: 1) against *data decryption*, strengthening ciphers and enabling privacy-preserving computation; and 2) against *watermark removal*, reinforcing embedding, detection, tamper localization/recovery, and proactive protection.

#### 2.4.1 Defenses against Data Decryption Attacks

Existing approaches fall into three broad classes: *(a) Chaos–neural hybrids*, which enlarge key space and resist statistical or differential attacks by combining chaotic maps with neural networks (Lakshmi et al., 2021);

*(b) Autoencoder-/GAN-based encryption*, which hide data via Cycle-GAN-based transformations that map images into hard-to-invert hidden domains (Ding et al., 2020); *(c) Privacy-preserving learning on encrypted data* using homomorphic encryption and secure multi-party computation, enabling distributed model training (Tang et al., 2019) and encrypted-domain inference (Bost et al., 2014) without ever exposing plaintext. These directions complement system hygiene against *side-channel leakage* and integrity risks (Benhamouda et al., 2018; Manikandan & Masilamani, 2018).

### 2.4.2 Defenses against Watermark Removal Attacks

We group defenses against Watermark Removal Attacks into four dimensions: *(a) Robust embedding* strengthens both visible and invisible watermark signals by integrating multiple embedding strategies. For visible marks, recent works employ multi-level alpha blending, adaptive texture-aware placement, and randomized geometric positioning to make watermark removal leave noticeable artifacts (Dekel et al., 2017). For invisible marks, robustness is achieved through transform-domain embedding (e.g., discrete cosine transform modification of frequency coefficients (Barni et al., 1998)), spread-spectrum coding with redundancy across channels (Cox et al., 1997), and synchronization patterns that maintain watermark alignment under rotation, scaling, or cropping (Lin & Chang, 1997). Together, these designs form the foundation of modern invisible and visible watermark protection (Wan et al., 2022).

*(b) Robust Detection and Verification* enhance the reliability of watermark verification through stronger statistical testing, improved scoring, and resilient encoding. For text and code watermarking, recent studies propose *finite-sample hypothesis tests* that avoid Gaussian approximations and enable more accurate detection under limited data conditions (Liu et al., 2024a; Yang et al., 2025b). Meanwhile, the work (Golowich & Moitra, 2024) analyzes the vulnerability of pseudo-random indexing watermarks under editing attacks, showing that robustness provably degrades with increasing edit distance. Together, these strategies form a layered defense, coupling operational safeguards with theoretical robustness for trustworthy watermark verification; *(c) Manipulation detection and content recovery*, which co-embed localization and self-recovery signals to make tampering both detectable and, when possible, reversible for image and document integrity, respectively (Ying et al., 2023; Cui et al., 2024); and *(d) Proactive data protection (Unlearnable Examples)*, which pre-perturbs data so that unauthorized training fails while preserving data utility, encompassing robust/stable (Liu et al., 2024c), semantic or feature-space perturbation defenses (Meng et al., 2024), transferable, model-free, and surrogate-free (Sadasivan et al., 2023), and cross-modal extensions with theoretical motivation (Jiang et al., 2024).

## 3 Data→Model (D→M)

Modern machine learning systems rely on large and diverse datasets to train foundation models. Because training pipelines may include data from untrusted or unverified sources, they present significant security and privacy risks. *D→M attacks* exploit this vulnerability by manipulating training, fine-tuning or inference data to implant malicious behaviors or bypass safety alignment. Unlike *D→D* attacks that modify the data itself, D→M attacks corrupt the learning or inference process, causing models to internalize unintended objectives and degrade in reliability. We focus on three representative families of D→M attacks: *data poisoning*, *harmful fine-tuning* and *jailbreak.*

The goal of *data poisoning attacks* is to corrupt the training data so that the model learns attacker-specified behaviors or incorrect objectives. Unlike test-time adversarial examples that perturb inputs after deployment, poisoning intervenes during model training or pre-training. Attackers modify samples, labels, or data flows so that empirical risk minimization optimizes a malicious objective, causing the resulting parameters to embed hidden biases or backdoors (Chen et al., 2017; Tian et al., 2022; Cinà et al., 2023). Such attacks can degrade model accuracy, trigger targeted misclassification, or implant covert functionality that persists even after alignment fine-tuning. They have been demonstrated across domains, from image and graph learning to LLM instruction tuning (Geiping et al., 2021), highlighting the vulnerability of model optimization in distributed and federated training settings.

Fine-tuning has become the standard mechanism for adapting foundation models to specific tasks. However, granting public or API-level access to fine-tuning pipelines introduces a new threat. The goal of *harmful fine-tuning attacks* is to manipulate the fine-tuning stage of foundation models to bypass safety alignment and induce undesired behaviors. Attackers upload small but malicious datasets, sometimes containing seemingly benign samples, to bias the model toward unsafe or target-specific responses. Unlike full retraining, these attacks require minimal data and computation yet can cause significant shifts in model behavior by exploiting the sensitivity of alignment layers. Recent studies (Qi et al., 2024) demonstrate that even minimal fine-tuning (using only a handful of carefully crafted examples) can drastically undermine alignment, suppress refusal behaviors, and enable the generation of restricted content, highlighting the fragility of current safety mechanisms.

The goal of *jailbreak attacks* (Liu et al., 2025a) is to bypass the safety alignment of large foundation models and trick them into producing harmful or restricted outputs. Unlike adversarial attacks that cause simple misclassification, jailbreaks directly override ethical or safe policy constraints. By breaking built-in safeguards, jailbreaks can lead to misinformation, malicious code, or other unauthorized behaviors.

## 3.1 Mathematical Formalization

A unified view of D→M attacks is that the adversary uses a model $f_{\boldsymbol{\theta}}$, a benign dataset $\mathcal{D}_{\text{clean}}$, together with a generation strategy $\text{G}(f_{\boldsymbol{\theta}}, \mathcal{D}_{\text{clean}})$ to produce a small set of malicious training or fine-tuning samples $\mathcal{D}_{\text{adv}}$, which are then injected into the benign dataset $\mathcal{D}_{\text{clean}}$ back. The goal is that the later resulting learning process yields parameters $\boldsymbol{\theta}$ that satisfy an attacker-specified objective while remaining stealthy on the model's normal tasks. This interaction can be formulated as a bilevel optimization:

$$\mathcal{D}_{\text{adv}}^* = G(f_{\boldsymbol{\theta}^*}, \mathcal{D}_{\text{adv}})$$
$$\text{s.t.} \quad \boldsymbol{\theta}^*(\mathcal{D}_{\text{clean}} \cup \mathcal{D}_{\text{adv}}) = \arg\min_{\boldsymbol{\theta}} \mathcal{L}_{\text{train}}(\boldsymbol{\theta}; \mathcal{D}_{\text{clean}} \cup \mathcal{D}_{\text{adv}})$$
$$S(\mathcal{D}_{\text{adv}}, \mathcal{D}_{\text{clean}}) \geq \tau.$$

$\mathcal{L}_{\text{train}}$ is the training (or fine-tuning) loss minimized on the dataset ($\mathcal{D}_{\text{clean}} \cup \mathcal{D}_{\text{adv}}$); it encodes the attacker's objective (e.g., increasing the rate of unsafe responses, forcing misclassification on triggered inputs, or reducing overall utility). The constraint $S$ is a similarity metric that expresses stealth (e.g., small cardinality, bounded perturbation, distributional similarity), $\tau$ is a quality threshold.

**Special Case I** The goal of **data poisoning attacks** (Tian et al., 2022) is to add or modify training samples so that the learned model parameters $\boldsymbol{\theta}^*$ produce attacker-specified failures (untargeted degradation) or targeted misbehaviour (backdoors). Concretely, the attacker constructs a poisoned dataset $\mathcal{D}_{\text{poison}}$ — generated via label manipulation, optimization-based perturbations, or related techniques — to optimize an adversarial objective $\mathcal{L}_{\text{adv}}$. Using the notation above, the poisoning optimization is:

$$\mathcal{D}_{\text{adv}}^* = \{(\boldsymbol{x}_j', y_j')\}_{j=1}^m = G(f_{\boldsymbol{\theta}^*}, \mathcal{D}_{\text{adv}})$$
$$G(f_{\boldsymbol{\theta}^*}, \mathcal{D}_{\text{adv}}) := \arg\min_{(\boldsymbol{x}', y')} \mathcal{L}_{\text{adv}}(\boldsymbol{\theta}^*(\boldsymbol{x}', y'); \mathcal{D}_{\text{target}})$$
$$\text{s.t.} \quad \boldsymbol{\theta}^*(\mathcal{D}_{\text{poison}}) = \arg\min_{\boldsymbol{\theta}} \mathcal{L}_{\text{train}}(\boldsymbol{\theta}; \mathcal{D}_{\text{poison}}),$$
$$\mathcal{D}_{\text{poison}} = \mathcal{D}_{\text{clean}} \cup \{(\boldsymbol{x}_j', y_j')\}_{j=1}^m, \|\boldsymbol{x}_j' - \boldsymbol{x}_j\|_p \leq \epsilon.$$

where := denotes definition, $(\boldsymbol{x}_j, y_j)$ denotes a clean training sample and $(\boldsymbol{x}_j', y_j')$ its poisoned counterpart, $y_j' \neq y_j$. $\mathcal{D}_{\text{target}}$ is a fixed set of target examples used to evaluate or trigger the adversarial objective (not an optimization variable), $m$ is the number of injected poisoning points, and $\epsilon$ controls the maximum perturbation magnitude under the $L_p$ norm.

*Untargeted Data Poisoning vs. Targeted Backdoor Poisoning.* Untargeted data poisoning aims to degrade a model's overall performance without enforcing any specific misbehavior. The attacker injects corrupted or mislabeled training samples so that the learned decision boundary becomes inaccurate across many inputs. The goal is broad degradation (such as reducing accuracy, increasing uncertainty) rather than controlling

how the model behaves on specific inputs. These attacks are often difficult to detect because the poisoned data may appear statistically similar to clean data, yet collectively distort the learning process.

In contrast, targeted backdoor poisoning is designed to induce specific, attacker-chosen behaviors while preserving normal performance on clean data. During training, the attacker injects a small number of poisoned samples containing a hidden trigger (e.g., a pattern, patch, or token) and assigns them a target label. After training, the model behaves normally on standard inputs but produces attacker-controlled outputs whenever the trigger is present.

**Special Case II** The goal of **harmful fine-tuning attacks** (Halawi et al.) is to compromise an aligned or pre-trained model by providing a small fine-tuning dataset $\mathcal{D}_{\text{adv}}$ (via API or shared service) so that the adapted parameters $\boldsymbol{\theta}_{\text{ft}}$ exhibit unsafe or biased behavior while retaining nearly unchanged performance on benign tasks, here $G$ constructs $\mathcal{D}_{\text{adv}}$ by following a set of predefined rules and procedures. The attack can be formulated as:

$$\mathcal{D}_{\text{adv}}^* = \{(\boldsymbol{x}_j, \boldsymbol{y}_j')\}_{j=1}^m = G(f_{\boldsymbol{\theta}^*}, \mathcal{D}_{\text{adv}}) := W(\{(\boldsymbol{x}_j, \boldsymbol{y}_j)\}_{j=1}^m)$$
$$\text{s.t.} \quad \boldsymbol{\theta}^* = \boldsymbol{\theta}_0,$$

Here W denotes the attacker's data-modification rules mapping a clean example $(\boldsymbol{x}, \boldsymbol{y})$ to a harmful example $(\boldsymbol{x}, \boldsymbol{y}')$. $\boldsymbol{\theta}_0$ are the original parameters. Typical attacker targets include removing refusal behavior, inserting triggers, or shifting outputs toward biased content. Such attacks can exploit parameter-efficient fine-tuning modules (e.g., LoRA adapters or prompt layers (Gao et al., 2024a)), often requiring only a few crafted examples and can be hard to detect.

**Special Case III** The goal of **jailbreak attacks** (Liu et al., 2025a) is to bypass the safety alignment of large foundation models, coercing them into generating harmful or restricted outputs by overriding built-in policy constraints rather than merely inducing prediction errors. A jailbreak attack can be formulated as a special case of the D→M paradigm in which the model parameters remain fixed, and the attacker instead manipulates input prompts to induce policy-violating behaviors at inference time. Let $f_{\boldsymbol{\theta}}$ denote a deployed (aligned) language model with fixed parameters $\boldsymbol{\theta}$, and let $\mathcal{D}_{\text{clean}}$ denote a set of benign user prompts. The attacker constructs adversarial prompts $\mathcal{D}_{\text{adv}}$ using a generation function $G$, such that

$$\mathcal{D}_{\text{adv}} = G(f_{\boldsymbol{\theta}}, \mathcal{D}_{\text{clean}}),$$

The attacker's objective is instead defined over the model's responses, aiming to induce policy-violating or unsafe outputs:

$$\max_{\mathcal{D}_{\text{adv}}} \quad \mathcal{L}_{adv}(f_{\boldsymbol{\theta}}; \mathcal{D}_{\text{adv}}) \quad \text{s.t.} \ S(\mathcal{D}_{\text{adv}}, \mathcal{D}_{\text{clean}}) \geq \tau,$$

where $S(\mathcal{D}_{\text{adv}}, \mathcal{D}_{\text{clean}}) \geq \tau$ is a stealth constraint, $S(\cdot, \cdot)$ measures semantic similarity, fluency, or perceptual closeness, and $\tau$ controls the degree of stealth. where $\mathcal{L}_{\text{adv}}$ measures jailbreak success, such as reducing refusal likelihood or increasing the probability of generating restricted content.

*Attacker Knowledge.* We use the term *detector* to denote the algorithm used to identify ownership signals, and the *key* refers to the hidden parameter or seed controlling embedding and decoding. The strength of an attack depends on the attacker's knowledge of the protection mechanism (Kirchenbauer et al., 2023). In a *black-box* setting, the adversary can only query a detector and observe binary outputs, with no access to the secret key or model internals. In a *gray-box* setting, partial information such as the architecture or general watermarking algorithm is available, but the secret key remains unknown. In a *white-box* setting, the attacker has full access to the detection system and can directly manipulate internal parameters to design optimized attacks.

Attack effectiveness is typically evaluated along two axes. First, *post-attack verifiability* measures the remaining strength of ownership or alignment constraints after the attack, i.e., the probability that the verification function $V$ still outputs 1 following transformation by $U$. Lower verifiability indicates more successful removal or circumvention of the protection mechanism. For example, in watermarking, this corresponds to post-attack detectability or decoding accuracy. Second, *utility preservation* assesses how much of the original data's perceptual, semantic, or functional quality is retained for downstream tasks. A successful attack achieves low verifiability while maintaining high utility, highlighting the inherent trade-off between stealth and fidelity.

### 3.2 Taxonomy and Techniques of Data→Model Attacks

In practice, data→model attacks fall into three broad families: *1) data poisoning* refers to the manipulation of training data such that standard learning procedures inadvertently optimize an adversarial objective; *2) harmful fine-tuning* uses small, carefully curated datasets to degrade a model's safety alignment or to implant malicious behaviors; and *3) jailbreak attacks*, which bypass alignment or safety rules in foundation models to force them to produce restricted or harmful content. In the following, we highlight the key taxonomic axes, common operational mechanisms, and representative works.

#### 3.2.1 Data Poisoning Attacks

Poisoning attacks manipulate training data to implant malicious behaviors or degrade model reliability. They can be systematically characterized along four orthogonal dimensions—*adversarial goals*, *label visibility*, *attacker knowledge*, and *training paradigm*—as discussed in recent comprehensive surveys (Tian et al., 2022; Nguyen et al., 2024b; Rodríguez-Barroso et al., 2023). Below, we summarize each axis and its representative techniques. *(a) Adversarial Goals.* Poisoning objectives are commonly divided into *availability attacks*, which reduce overall model utility (e.g., accuracy or calibration), and *integrity attacks*, which implant targeted behaviors such as backdoors or hidden triggers that activate only under specific conditions (Gu et al., 2017). Recent work further identifies more subtle objectives, such as degrading model confidence or calibration without label manipulation (Chaalan et al., 2024). *(b) Label Visibility.* Depending on whether the attacker manipulates labels, poisoning can be either *dirty-label* or *clean-label*. Dirty-label attacks explicitly flip or corrupt training labels, whereas clean-label attacks preserve ground-truth labels but perturb inputs to mislead the learned decision boundary—typically by generating adversarial-like examples that induce feature collisions (Shafahi et al., 2018). *(c) Attacker Knowledge.* The strength and strategy of poisoning attacks vary with the attacker's access level. In *white-box* settings, full access to model parameters and gradients enables optimization-based attacks, typically formulated as bilevel optimization and often approximated through influence functions (Koh et al., 2022). *Gray-box* attackers possess partial information (such as model architecture or aggregation rules in federated learning) and adapt their poisons accordingly (Rodríguez-Barroso et al., 2023). In contrast, *black-box* attackers lack internal access but can still exploit transferability from surrogate models (Zhu et al., 2019), demonstrating that effective poisoning remains feasible even without visibility into gradients or training data (Chen et al., 2023a; Liu & Lai, 2021). *(d) Training Paradigm.* Poisoning manifests differently across learning paradigms. In conventional centralized training, attackers can directly manipulate datasets by injecting crafted samples or flipping labels (Ramirez et al., 2022). In federated learning, poisoning (Xie et al.) arises through malicious client updates or model replacement, where even a single compromised participant can bias the global aggregation and degrade overall model integrity (Nguyen et al., 2024b). Similar paradigm-specific threats extend beyond vision to other modalities—graphs (node or edge injection) (Cinà et al., 2023; Tao et al., 2021), time series (temporal or phase triggers) (Lin et al., 2024), and language (rare-token, syntactic, or instruction-pattern triggers that persist through fine-tuning) (Kurita et al., 2020). At the scale of foundation models, even tiny poisoning ratios can survive model-level safety alignment and propagate through subsequent updates (Bowen et al., 2025; Carlini et al., 2024a), highlighting the need for domain-aware validation pipelines in high-stakes applications such as biomedical large language models (Alber et al., 2025).

#### 3.2.2 Harmful Fine-tuning Attacks

Harmful fine-tuning manipulates the adaptation stage of aligned models using small curated datasets, leading to unsafe or biased behaviours while retaining benign utility. Existing studies reveal several recurring patterns, summarized below. *(a) Malicious Data Poisoning.* Attackers deliberately insert adversarial prompt–response pairs into the fine-tuning dataset to overwrite safety alignment. As demonstrated in work (Yi et al., 2024a), applying *reverse alignment*—either via Reverse Supervised Fine-Tuning or Reverse Preference Optimization—can fine-tune safety-aligned open-access LLMs to undermine refusal behaviors and weaken built-in safeguards. *(b) Benign-Data–Induced Misalignment.* Even datasets without explicit harmful content can degrade safety: outlier yet non-toxic examples, distributional biases, or latent correlations in seemingly benign corpora may erode alignment to a degree comparable with adversarial fine-tuning (He et al.). These results highlight that alignment failure can arise naturally from poor data curation rather

than from deliberate attacks. *(c) Parameter-Efficient and Model-Specific Pathways.* Attackers exploit architectural properties and lightweight adaptation mechanisms to inject harmful logic efficiently. Single-edit or adapter-based backdoors, targeted knowledge editing, and PEFT-as-attack demonstrate how small parameter updates can trigger disproportionate behavioral changes (Chen et al.). *(d) API Exploitation and Service Abuse.* In hosted environments, attackers exploit fine-tuning APIs to upload covert malicious data. Recent work (Halawi et al.) shows that pointwise-undetectable datasets, mixed compliance–refusal fine-tuning objectives, and constrained black-box jailbreaks can degrade safety even without access to model weights. Related findings indicate that open fine-tuning interfaces remain an inherent security risk for both commercial and open-source LLMs. Beyond these categories, recent studies reveal that harmful fine-tuning can also emerge in more subtle forms. Emergent misalignment may appear during narrow-domain or agentic fine-tuning, where self-updating models gradually drift toward unsafe policies (Hahm et al., 2026; Wallace et al., 2025; Shao et al., 2025b). Together, these observations show that even small, seemingly benign datasets (whether maliciously designed or inadvertently mis-specified) can reliably steer large foundation models toward unsafe or biased behaviors, underscoring the fragility of current alignment processes.

### 3.2.3   Jailbreak Attacks

Jailbreaks bypass alignment or safety mechanisms to force models to output restricted or harmful content that would normally be blocked, posing a major threat to LLMs and multimodal models. They can be characterized by: *(a) attacker access*, distinguishing white-box gradient-guided suffix generation from black-box prompt rewriting and role-play prompting (Geisler et al.); *(b) attack timing*, including training-stage backdoor injection and inference-stage adversarial prompting (Chao et al., 2024); *(c) prompt manipulation strategies*, such as template completion, scenario nesting, cipher-based rewriting, or genetic optimization (Chao et al., 2024); and *(d) system-level extensions*, where multimodal or agentic systems are compromised through adversarial cross-modal cues or external tools/APIs/ memory manipulation for autonomous agents (Xu et al., 2024b; Liu et al., 2025a). Subsequent studies have expanded jailbreak analyses to multimodal systems. (Dong et al., 2023) systematically evaluate adversarial image attacks against Google's Bard, revealing that small, imperceptible perturbations can bypass both face- and toxicity-detection modules, thereby exposing critical vulnerabilities in visual-language alignment and safety filtering of commercial MLLMs. More recently, research has shown that such system-level vulnerabilities also extend to agentic architectures. For example, (Li et al., 2025a) demonstrate that even commercial LLM-based web and scientific agents are vulnerable to trivial yet dangerous prompt-injection and redirection attacks. These attacks demonstrate how behavioral safeguards can be circumvented even without modifying model parameters.

## 3.3   Defensive Mechanisms

### 3.3.1   Defense against Data Poisoning

Defenses against data poisoning aim to preserve model integrity despite the presence of malicious or corrupted data. They can be broadly organized by where the intervention occurs: *during training* or *at inference.* Across all settings, effective defense combines anomaly detection, robust optimization, and trust-aware data filtering to limit the adversary's impact.

*(a) Training-Time Defenses.* Training-stage defenses focus on identifying or neutralizing poisoned data before or during optimization. Data-level regularization via *augmentation* (e.g., Mixup, CutMix) helps dilute backdoor triggers and reduces memorization of malicious patterns (Borgnia et al., 2021). At the loss-function level, *robust learning objectives* derived from noisy-label and meta-learning literature—such as ITLM (Iterative Trimmed Loss Minimization) (Shen & Sanghavi, 2019), GCE (Zhang & Sabuncu, 2018), reweighting-based methods (Ren et al., 2018), Co-teaching (Han et al., 2018), and MentorNet (Jiang et al., 2018)—limit the influence of high-loss or inconsistent samples. Feature-space filtering methods like *De-Pois* (Chen et al., 2021a) further cluster examples by representation consistency to remove those with mismatched feature–label semantics. A framework *Neural Attention Distillation (NAD)* (Li et al.) was proposed to use a finetuned teacher network to guide a backdoored student model via attention alignment on a small clean subset. Building on this line, *Li et al.* (Li et al., 2021b) introduced *Anti-Backdoor Learning (ABL)*, a training-time paradigm that aims to train clean models directly on poisoned data. However, most training-time defenses

remain largely ineffective against *clean-label* or *stealthy backdoor* attacks, in which poisoned samples appear statistically benign and evade standard anomaly detectors (Koh et al., 2022; Geiping et al., 2021).

*(b) Inference-Time Defenses.* Inference-time defenses operate after model deployment and aim to detect or mitigate triggered behavior at test time. A first line of defense is *anomaly detection*, which removes outlier samples using statistical or representation-based criteria such as clustering, spectral signatures, or isolation forests (Cinà et al., 2023). *Uncertainty-based filtering* evaluates prediction entropy or stability under perturbations: low-entropy or invariant outputs often reveal the presence of triggers. Typical examples include STRIP for vision (Gao et al., 2019), which uses entropy-based uncertainty signals to identify poisoned inputs. Other work (Liu et al., 2023) tests robustness under input corruptions (noise, blur, occlusion), exploiting the observation that poisoned examples remain unusually stable under such changes. A complementary direction is *knowledge-guided validation*, which cross-checks model predictions against external knowledge sources, such as biomedical knowledge graphs, to flag implausible outputs (Alber et al., 2025).

### 3.3.2 Defense against Harmful Fine-tuning

Defenses against harmful fine-tuning aim to preserve model safety and alignment even when adversaries attempt to retrain or adapt models with malicious or misleading data. Existing methods can be broadly grouped into three complementary directions. *(a) Alignment-Stage Immunization.* These approaches fortify models during the initial alignment phase to make them intrinsically resistant to future harmful fine-tuning. Representative methods enhance weight stability, representation invariance, or regularization against adversarial updates—such as perturbation-aware and layer-wise robustness training (*Vaccine* (Huang et al., 2024c), *T-Vaccine* (Liu et al., 2025b)), loss-based regularization and proximal optimization (*Booster* (Huang et al., 2025c), *LISA* (Huang et al., 2024a)), and representation-level noise injection (*RepNoise* (Rosati et al., 2024b)). Formal analyses further define theoretical *immunization conditions* (including resistance, stability, and generalization) that guide preventive alignment strategies (Rosati et al., 2024a). *(b) In-Training Safeguards.* These defenses are applied at fine-tuning time to monitor and mitigate malicious model updates in real time. *Self-Degraded Defense (SDD)* (Chen et al., 2025c) pre-emptively trains models by pairing harmful prompts with benign, high-quality responses, thereby reducing the model's sensitivity to malicious data while preserving its normal capabilities. For parameter-efficient fine-tuning (PEFT), *PEFTGuard* (Sun et al., 2025) detects backdoored adapters by directly transforming and classifying their weight tensors (e.g., LoRA) with a parameter-only meta-classifier, and identifying backdoor-specific patterns with near-perfect accuracy across tasks. Bayesian data scheduler (Hu et al., 2025) incorporates probabilistic safety control by assigning weights to samples according to their posterior safety attributes during fine-tuning, effectively suppressing the influence of unsafe or malicious data on model adaptation. *(c) Post-Tuning Repair.* When harmful fine-tuning has already occurred, post-hoc repair methods attempt to recover safety without retraining from scratch. *Antidote* (Huang et al., 2025a) prunes harmful parameters identified via importance scoring, effectively restoring alignment with minimal loss in utility. Such an approach treats fine-tuning as reversible damage, focusing on repairing rather than preventing misalignment.

### 3.3.3 Defenses against Jailbreak Attacks

We organize jailbreak attack defenses into three main categories based on the protection level: *(a) Prompt-level* screening and rewriting—detecting risky or injected inputs via fine-tuned classifiers and heuristic filters, and sanitizing or rewriting them before the model processes the prompt (Jacob et al., 2024); *(b) Model-level* alignment and steering—reinforcement learning from human feedback (RLHF) (Ouyang et al., 2022) and related safety fine-tuning enhance model alignment and refusal behavior, while decoding-time constraints and internal-signal detectors further mitigate unsafe generations (Ouyang et al., 2022); and *(c) System-level* guardrails—benchmark-driven guard models, multi-stage filters for multimodal LLMs, tool/memory governance for agents, and continuous runtime monitoring at deployment (Huang et al., 2024d; Chao et al., 2024). Recent advances expand this line of work: *AIR-BENCH 2024* (Zeng et al.) introduces a regulation-aligned auditing framework that evaluates model refusal behaviors and compliance across real-world risk categories; *T2VSafetyBench* (Miao et al., 2024) extends evaluation to text-to-video generative models, revealing multimodal jailbreak vulnerabilities; and *AEGIS-LLM* (Cai et al.) demonstrates that incorporating auxiliary agent roles and leveraging automated prompt optimization can enhance system robustness without com-

promising task utility. In practice, these defenses must account for diverse attack settings—including both white- and black-box access, and attacks occurring at training or inference time—such as suffix optimization, backdoor injection, role-play or cipher-based prompt rewriting, and evolutionary prompt search (Liu et al., 2025a).

# 4   Model→Data (M→D)

Attacks in the Model→Data direction aim to infer the information that a trained model implicitly encodes about its training data. Rather than stealing parameters or manipulating the model externally, these attacks exploit what the model *memorizes*—its ability to reveal, reconstruct, or statistically expose private training data set samples. Such leakage undermines data confidentiality and consent, as even deployed or API-restricted models may inadvertently disclose sensitive content through their outputs, embeddings, or confidence patterns. Within this category, we highlight three representative families of privacy-violating attacks: *model inversion attacks*, *membership inference attacks*, and *training data extraction attacks.*

The goal of *model inversion attacks* (Fredrikson et al., 2015; Yang et al., 2025a; Dibbo, 2023) is to reconstruct sensitive information about the training data directly from a trained model. By exploiting confidence scores, embeddings, or gradients, an adversary can approximate original data features or even recover realistic samples such as faces or text segments. Early studies assumed white-box access, but recent work shows that inversion can succeed in black-box APIs by leveraging systematic output patterns. These attacks reveal how models encode detailed traces of their training data even without direct access to the dataset itself.

The goal of *membership inference attacks* is to determine whether a specific data instance (or an entire subset) was used in a model's training process. By probing the model and analyzing statistical differences between "seen" and "unseen" samples—often manifested in confidence scores, loss values, or hidden representations—attackers can infer a data's participation in sensitive datasets such as medical or personal records. Recent research extends these attacks beyond conventional classifiers to LLMs and diffusion models, where memorization and overfitting amplify data membership signals.

The goal of *training data extraction attacks* is to induce a model to directly reproduce fragments of its original training data, rather than merely inferring or reconstructing them statistically. Through carefully crafted prompts, triggers, or fine-tuning procedures, adversaries can compel LLMs or diffusion models to regenerate exact text passages, images, or identifiers memorized during pre-training. Unlike inversion or membership inference, these attacks cause the model to emit verbatim training content, posing severe risks to privacy, copyright, and regulatory compliance in generative systems.

## 4.1   Mathematical Formalization

Model→Data attacks exploit information memorized within trained models to recover or expose private training data. Given a model $f_{\boldsymbol{\theta}}$ trained on $\mathcal{D}_{\text{train}}$, the objective $G$ of the attacker is to extract training data set information $\boldsymbol{x}_{\text{info}}$ from the outputs or representations of the model $f_{\boldsymbol{\theta}}$:

$$\boldsymbol{x}_{\text{info}} = G(f_{\boldsymbol{\theta}}), \quad \text{s.t.} \quad \boldsymbol{x}_{\text{info}} \sim \mathcal{T}(\mathcal{D}_{\text{train}})$$

Here $\mathcal{T}(\mathcal{D}_{\text{train}})$ denotes the information about the training dataset that the attacker seeks to recover or infer from the model (e.g., specific samples, attributes, or membership signals). Constraints capture model access level and query limits. Different instantiations of $G$ yield three major M→D families: *model inversion attack*, *membership inference attack*, and *training data extraction attack.*

**Special Case I   Model Inversion Attacks** (Yang et al., 2025a). At a high level, model inversion attacks attempt to *run the model backward.* Instead of using a trained model to predict an output from a given input, the attacker starts from a desired output (e.g., a class label or prediction) and tries to recover an input that would produce that output under the model. The attacker treats the input $\boldsymbol{x}$ itself as an optimization variable and searches for a reconstructed sample $\hat{\boldsymbol{x}}$ whose prediction under the model matches the target output. Under such attacks, $\boldsymbol{x}_{\text{info}}$ is the reconstructed data $\hat{\boldsymbol{x}}$ that aligns with the distribution of $\mathcal{D}_{\text{train}}$.

Given a target output $y^*$, this is done by minimizing an inversion objective that balances two goals and the attacker defines an inversion objective function $G$ as:

$$\hat{\boldsymbol{x}} = G(f_{\boldsymbol{\theta}}) := \arg\min_{\boldsymbol{x}} \left[ \mathcal{L}_{\mathrm{inv}}(f_{\boldsymbol{\theta}}(\boldsymbol{x}), y^*) + \lambda \, \mathcal{R}(\boldsymbol{x}) \right],$$

$$\text{s.t.} \quad \hat{\boldsymbol{x}} \sim \mathcal{T}(\mathcal{D}_{\mathrm{train}}).$$

where := denotes definition, $\mathcal{L}_{\mathrm{inv}}$ enforces output consistency with the target $y^*$, and $\mathcal{R}$ regularizes the realism of reconstructed samples. Both white-box (gradient-based) and black-box (API-query) settings can reveal sensitive attributes or even approximate original training samples. Here, $y^*$ denotes the target output (label or response) with respect to which the attacker seeks an input $\hat{\boldsymbol{x}}$ whose model prediction matches $y^*$, and $\mathcal{T}(\mathcal{D}_{\mathrm{train}})$ represents the distribution of the training dataset.

**Special Case II  Membership Inference Attacks (MIA)** (Hu et al., 2022a) determine a a binary membership signal $\boldsymbol{x}_{\mathrm{info}}$ whether a specific data sample $\boldsymbol{x}^*$ was part of the training dataset $\mathcal{D}_{\mathrm{train}}$. Formally, the attacker designs a discriminator $g$ (or classifier) that takes the model $f_{\boldsymbol{\theta}}$ outputs or representations to predict whether $\boldsymbol{x}^*$ belongs to $\mathcal{D}_{\mathrm{train}}$. The objective of $G$ can be written as:

$$g^* = G(f_{\boldsymbol{\theta}}) := \arg\min_{g} \; \mathcal{L}_{\mathrm{mem}}\big(g(f_{\boldsymbol{\theta}}(\boldsymbol{x}^*)), \, m^*\big),$$

$$\text{s.t.} \quad g(f_{\boldsymbol{\theta}}(\boldsymbol{x}^*)) \in [0, 1].$$

where $\mathcal{L}_{\mathrm{mem}}$ denotes the membership classification loss (e.g., binary cross-entropy), and $m^* \in \{0, 1\}$ is the ground-truth membership label indicating whether the target sample $\boldsymbol{x}^*$ belongs to the training dataset. The trained discriminator outputs $g(f_{\boldsymbol{\theta}}(\boldsymbol{x}^*))$ lies in $[0, 1]$ and is defined with respect to the training-data information $\mathcal{T}(\mathcal{D}_{\mathrm{train}})$.

**Special Case III  Training Data Extraction Attacks** (Xu et al., 2024a) aim to induce a generative model to directly output training samples or fragments of them. Unlike model inversion attacks, which reconstruct inputs by solving an optimization problem, training data extraction attacks rely on repeatedly querying the model with carefully chosen prompts or sampling strategies. The attacker does not modify the model parameters, but instead interacts with the model in a way that increases the likelihood that its generated outputs closely match examples from the training data.

Given a generative model $f_{\boldsymbol{\theta}}$ that defines a conditional distribution $p_{\boldsymbol{\theta}}(\boldsymbol{x} \mid \boldsymbol{q})$ over outputs given a query or prompt $\boldsymbol{q}$, the attacker constructs an extraction operator $E$ that interacts with $f_{\boldsymbol{\theta}}$ to recover samples consistent with the training-data information $\mathcal{T}(\mathcal{D}_{\mathrm{train}})$. In this case, $\boldsymbol{x}_{\mathrm{info}}$ corresponds to the generated samples that the attacker extracts from the model, in this case, denotes as $\tilde{\boldsymbol{x}}$, which are expected to align with the training data distribution $\mathcal{T}(\mathcal{D}_{\mathrm{train}})$. The objective of $G$ can be expressed as:

$$E^* = G(f_{\boldsymbol{\theta}}) := \arg\min_{E} \; \mathbb{E}_{\tilde{\boldsymbol{x}} \sim f_{\boldsymbol{\theta}}(\cdot \mid E)}\big[\mathcal{L}_{\mathrm{rec}}\big(\tilde{\boldsymbol{x}}, \, \mathcal{T}(\mathcal{D}_{\mathrm{train}})\big)\big],$$

$$\text{s.t.} \quad \tilde{\boldsymbol{x}} = f_{\boldsymbol{\theta}}(\cdot \mid E^*) \approx \mathcal{T}(\mathcal{D}_{\mathrm{train}}).$$

Here, $\approx$ denotes approximately equal to, $E$ denotes the attacker's extraction operator (e.g., a query generator, decoding policy, or sampling strategy) that issues prompts or queries to $f_{\boldsymbol{\theta}}$. Given a model $f_{\boldsymbol{\theta}}$, the attacker can obtain a $\tilde{\boldsymbol{x}}$ through $E$, which is approximately equal to certain information contained in $\mathcal{T}(\mathcal{D}_{\mathrm{train}})$; $\mathcal{L}_{\mathrm{rec}}$ measures reconstruction fidelity or semantic similarity between generated outputs $\tilde{\boldsymbol{x}}$ and the target training information $\mathcal{T}(\mathcal{D}_{\mathrm{train}})$. In practice, $E$ may operate *targetedly*—optimizing queries toward a specific sample $\boldsymbol{x}^*$ or identifying memorized content associated with known attributes—or *untargetedly*, by probing the model to elicit any memorized fragments through repeated sampling.

*Access assumptions* The signals available to the adversary vary with the threat model: *(a) White-box access (Fredrikson et al., 2015):* full gradients or hidden activations can be exploited to optimize reconstructions directly. *(b) Black-box access:* the adversary can only interact with the model through its outputs. Two common variants are: (b.1) probability access (Zhang et al., 2020), where softmax scores or logits are returned and provide richer information for inversion; and (b.2) label-only (Kahla et al., 2022), where only the top-1 predicted class is visible, making query efficiency critical.

### 4.2 Taxonomy and Techniques of Model→Data

#### 4.2.1 Model Inversion Attacks

Model inversion attacks (MIAs) aim to reconstruct sensitive training data or its representative features from a model's accessible information, such as outputs, gradients, or embeddings. Recent studies (Dibbo, 2023; Yang et al., 2025a) have established MIAs as one of the core privacy threats linking models back to their data, with diverse forms depending on the attacker's objective, access, and prior knowledge. Below we outline these key dimensions and the main technical paradigms observed across modalities. *(a) Target Type.* Depending on the reconstruction granularity, attacks may operate at the *instance level*, recovering individual samples (Fredrikson et al., 2015; 2014); the *class level*, reconstructing category prototypes or feature representations (Hitaj et al., 2017); or the *distribution level*, approximating the overall data manifold through semantic or statistical priors (Chen et al., 2021b). *(b) Access Level.* Depending on access, white-box settings reveal gradients or hidden activations (Zhang et al., 2020; Hu et al., 2023a; Wei et al., 2024b;a), black-box settings expose only logits or labels (Kahla et al., 2022; Hu et al., 2023b), and mixed cases exploit side channels or shared embeddings (Chanpuriya et al., 2021). *(c) Prior Knowledge.* Attackers may leverage auxiliary in-domain information memorized by the model itself (Carlini et al., 2019), or employ synthetic priors—such as noise-based reconstructions or randomly generated queries—to approximate the target label data distribution (Truong et al., 2021; Tramèr et al., 2016).

Across various settings, methods can be broadly categorized into five families. (a) *Optimization-based inversion* reconstructs inputs by maximizing confidence or minimizing feature discrepancy on target labels, often regularized by perceptual or total-variation priors (Zhang et al., 2020; Mahendran & Vedaldi, 2015). (b) *Learning-based inversion* further trains up-convolutional networks to directly map feature representations back to images (Dosovitskiy & Brox, 2016). (c) *Generative inversion* (Wang et al., 2021a) employs conditional GANs or variational inference to sample plausible reconstructions. (d) *Representation-space inversion* (Tragoudaras et al., 2025) decodes intermediate embeddings into the input domain, while (e) *prompt- or explanation-guided inversion* (Morris et al., 2024; Zhao et al., 2021) leverages logits, gradients, or attribution maps to refine reconstruction quality.

While early work focused on vision, similar mechanisms now extend to text (Morris et al., 2024), graphs (Zhou et al., 2023), time series, and medical signals (Subbanna et al., 2021; Ghimire et al., 2018). Diffusion models exhibit both data extraction (Carlini et al., 2023) and prompt-level inversion behaviors (Mahajan et al., 2024). At the foundation scale, increasing capacity amplifies memorization (Carlini et al., 2021), enabling instance-level leakage even through limited-access interfaces (Dibbo, 2023; Carlini et al., 2022). Model inversion can target single samples, classes, or distributions, exploiting gradients, logits, or representations to reverse the data–model mapping (Dibbo, 2023; Wei et al., 2025).

#### 4.2.2 Membership Inference Attacks

Membership inference attacks aim to determine whether a particular sample—or a collection of samples—was included in a model's training data set. By exploiting prediction behavior, hidden activations, or gradients, these attacks expose whether the model *memorize* specific data, revealing data participation information and thus breaching data privacy. Current approaches can be broadly grouped into four families: (a) output-based black-box attacks, (b) internal-signal and white-box attacks, (c) data-extraction–driven leakage, and (d) domain- or system-specific extensions. *(a) output-based black-box attacks.* These methods rely only on model outputs such as likelihoods, confidence scores, or generated text. Prompt- and perturbation-based attacks (Fu et al., 2025) probe stability in next-token probabilities or generations to separate members from non-members. Likelihood-based detection at the *dataset/corpus* level is re-evaluated and formalized by a new inference method (Maini et al., 2024). In retrieval-augmented generation (RAG) systems, membership can be inferred from the semantic similarity and perplexity between retrieved knowledge and generated text, revealing private entries in the external database (Li et al., 2025c). Complementarily, black-box provenance detection frameworks for LLMs—e.g., *DPDLLM*—identify whether text likely appeared in pre-training without logit access (Zhou et al., 2024). *(b) Internal-Signal and White-Box Attacks.* When gradients, weights, or activations are observable, stronger inference becomes possible. Gradient- and parameter-based methods (Pang et al., 2023; Suri et al., 2024) leverage differential signals to expose training membership, while

neuron-level attribution analysis (*Unveiling the Unseen*) (Li et al., 2024a) identifies internal activations correlated with membership cues. Statistical testing frameworks such as *Low-Cost High-Power Membership Inference Attacks* (RMIA) (Zarifzadeh et al., 2024) further improve sensitivity and robustness under limited reference models. For large language models, *memTrace* (Makhija et al., 2025) extracts membership signals from hidden-state dynamics and attention patterns. At the other end of the access spectrum, *OSLO* (Peng et al., 2024) demonstrates that even label-only interfaces enable high-precision inference via transfer-based adversarial perturbations. Finally, explainability mechanisms themselves can act as side channels, as attribution maps and confidence changes from explanators reveal membership information (Liu et al., 2024b). *(c) Data extraction and inversion leakage.* These attacks reveal a continuum between membership inference and explicit data reconstruction. Diffusion and language models can directly regenerate training samples through repeated prompting or sampling (Carlini et al., 2023; 2021), showing that memorization and membership inference are deeply entangled. *(d) Domain-specific and system-level extensions.* Beyond standard classifiers, membership inference has been explored in graph contrastive learning, recommender systems, and biometric recognition (Wang & Wang, 2024; Zhong et al., 2024). Comprehensive evaluations (DeAlcala et al., 2024; Zhu et al., 2024a) identify key influencing factors across centralized and federated settings, while information-theoretic and learning-based calibration analyses (Zhu et al., 2025b; Shi et al., 2024) provide finer-grained quantification of leakage. Membership inference spans a continuum from black-box querying to gradient-level forensics, connecting with data extraction, inversion, and unlearning analysis. Across modalities and model scales, these studies collectively show that memorization remains a fundamental and quantifiable privacy risk in modern machine learning.

### 4.2.3 Training Data Extraction Attacks

Training data extraction aims to recover verbatim training data from deployed models and can be grouped by the adversary's access level and manipulation capability. Existing work mainly focuses on *query-based extraction*, which elicits memorized content via prompt engineering and sampling artifacts. Black-box adversaries craft prompts or exploit sampling errors to trigger memorized responses. Sequence-level studies (Xu et al., 2024a) show that shorter prefixes and larger models tend to leak more. Recent work by Nasr et al. (Nasr et al., 2025) introduces two scalable attacks (*divergence* and *finetuning*) that enable large-scale recovery of proprietary training data even under restricted, publicly accessible interfaces. Building on this direction, More et al. (More et al., 2025) examine more realistic adversarial settings, showing that prompt sensitivity, access to multiple checkpoints, and downstream tasks can amplify extraction risks, revealing a stronger composite adversary that better captures real-world threat conditions. Parallel efforts extend these attacks to generative diffusion models: Carlini et al. (Carlini et al., 2023) demonstrate training-image extraction from diffusion models.

### 4.3 Defensive Techniques.

### 4.3.1 Defenses against Model Inversion Attacks

These defenses aim to protect the privacy and confidentiality of the data used to train or query a model. They can be broadly categorized into three complementary directions: *(a) Training-Time Privacy Regularization.* The dominant approach is differentially private stochastic gradient descent (DP-SGD) (Abadi et al., 2016), which injects calibrated noise into gradient updates to bound each sample's contribution. While DP provides formal privacy guarantees, it often degrades accuracy. Complementary strategies introduce implicit regularization: information-bottleneck–inspired methods such as bilateral dependency optimization (Peng et al., 2022) constrain representations to retain only task-relevant features, and stochastic mechanisms such as dropout (Srivastava et al., 2014) mitigate overfitting and implicitly reduce memorization. *(b) Inference-Time Output Obfuscation.* Since most inversion attacks rely on observable outputs, these defenses modify predictions to conceal exploitable signals. Label smoothing (Müller et al., 2019) reduces confidence gaps across classes and improves calibration, while adversarial regularization (Wen et al., 2021) or randomized post-processing of logits weakens gradient-based reconstruction cues. *(c) Post-Deployment Detection and Perturbation Frameworks.* Runtime defenses focus on detecting inversion-like queries or embedding irreversible transformations into model internals. Semantic perturbation–based frameworks (Zhu et al., 2024b)

analyze query embeddings and behavioral consistency, introducing statistical signatures that help distinguish or distort inversion-derived surrogates.

### 4.3.2 Defenses against Membership Inference Attacks

These defenses aim to eliminate the statistical gap between members and non-members observable from model outputs, gradients, or representations. Existing studies fall into four major categories: *(a) unlearning and token-level mitigation*, *(b) ensemble and distillation strategies*, *(c) noise injection and regularization*, and *(d) generative and adversarial training*.

*(a) Unlearning and Token-Level Mitigation.* Selective unlearning treats memorized content differently from general knowledge. *Tokens for Learning, Tokens for Unlearning* (Tran et al., 2025) jointly optimizes learning and unlearning objectives by categorizing tokens into *hard and memorized*, reducing membership leakage with minimal impact on language modeling performance. Other unlearning methods apply targeted forgetting or data editing to erase memorized content while preserving task utility. *(b) Ensemble and Distillation Strategies.* These methods aggregate or transfer knowledge across multiple models to dilute membership signals. Multi-teacher and repeated distillation frameworks (Zheng et al., 2021; Shejwalkar & Houmansadr, 2021) transfer softened or masked predictions from teacher models to students, mitigating overconfident behaviors and enabling tunable privacy–utility trade-offs. Ensemble-based defenses such as MIAShield (Jarin & Eshete, 2023) mitigate membership inference by preemptively excluding models trained on the queried sample, thereby eliminating strong membership signals while preserving utility. *(c) Noise Injection and Regularization.* A widely used direction is to perturb training or inference signals to blur member/non-member distinctions. Noise injection methods (e.g., Weighted Smoothing (Tan et al., 2023)) adaptively add perturbations to high-risk samples, while regularization-based defenses (e.g., MIST (Li et al., 2024b) and NeuGuard (Xu et al., 2022)) constrain model representations or neuron activations to reduce membership inference vulnerability. Graph perturbation (Wang et al., 2023a) obscures membership signals by injecting noise into graph structures, while enhanced Mixup (Chen et al., 2021c) and weight pruning (Wang et al., 2021b) regularize models to reduce overfitting and memorization. *(d) Generative and Adversarial Training.* Generative defenses leverage GAN- or VAE-based frameworks to generate synthetic data to train or regularize generative models, in order to obscure membership signals while preserving utility (Hu et al., 2022b; Mukherjee et al., 2021; Yang et al., 2023b). Digestive neural networks (Lee et al., 2021) sanitize shared gradients in federated settings, and adversarial regularization (Nasr et al., 2018) jointly trains a classifier and adversary to produce membership-resistant representations.

### 4.3.3 Defenses against Training Data Extraction Attacks

Large language and diffusion models may memorize and expose sensitive training examples through overfitting or sampling. Defenses, therefore, aim to suppress memorization, distort membership signals, or detect exposed content. Existing works can be grouped into four complementary directions. *(a) Training-Time Regularization and Noise.* These defenses modify the optimization process to ensure similar behavior between members and non-members. Differentially private fine-tuning (e.g., DP-SGD or DP-Adam) (Du et al., 2025) provides formal privacy guarantees against data leakage and, when combined with low-rank adaptation (LoRA), achieves a favorable privacy–utility trade-off. *(b) Architectural and Ensemble Isolation.* Re-architecting models to decouple knowledge across data subsets prevents any single component from over-memorizing, as demonstrated by SELENA (Tang et al., 2022), which trains multiple sub-models on overlapping subsets and uses self-distillation to align their behaviors. *(c) Query- and Output-Level Perturbation.* Post-training defenses perturb model queries or responses to obfuscate membership signals. QUEEN (Chen et al., 2025a) adaptively perturbs sensitive queries and reverses gradients to corrupt extraction attempts in model-stealing scenarios. Beyond query perturbation, output watermarking offers another form of post-generation modification. Panaitescu et al. (Panaitescu-Liess et al., 2025) show that output watermarking can significantly reduce the probability of verbatim memorization, thereby preventing copyrighted text generation. *(d) Memorization Detection and Auditing.* Rather than suppressing leakage, these defenses identify and monitor it. Diffusion-model audits (Wen et al., 2024a) detect memorized samples via prompt-conditioned prediction analysis, while LLM auditing frameworks such as ContextLeak (Choi et al.) insert canaries or triggers to trace data exposure during fine-tuning and in-context learning.

# 5 Model→Model (M→M)

## 5.1 Model Extraction Attacks

Model Extraction Attacks (MEAs) (Liang et al., 2024) pose a critical threat to the confidentiality and intellectual property (IP) of machine learning models, particularly in the context of Machine Learning as a Service (MLaaS) (Kesarwani et al., 2018). Model extraction attacks aim to clone a deployed machine-learning model by repeatedly querying it and observing its outputs. An attacker does not need access to the model's parameters or training data, only the ability to send inputs (e.g., images or text) and receive predictions. By carefully choosing queries and collecting the corresponding outputs, the attacker can train a new *surrogate* model that closely mimics the behavior of the original one. This surrogate can approximate not only the model's decision boundaries, but sometimes its architecture or parameters, undermining commercial value and IP protection. Over time, this stolen model can achieve similar accuracy and functionality, effectively reproducing the victim's intellectual property. Model extraction is particularly concerning because it enables downstream misuse: the extracted model can be analyzed offline, fine-tuned for harmful purposes, or used to launch further attacks such as model inversion or data leakage, all while bypassing access controls and usage limits of the original system.

MEAs can be broadly classified into two main categories: *(1) Functionality Stealing (Truong et al., 2021)*: the adversary aims to replicate the prediction performance of the victim model, producing a substitute that yields consistent outputs. Depending on the availability of data, functionality stealing can be further divided into: *(a) Data-based Model Extraction (DBME)*, where attackers leverage knowledge of the training dataset used for training the target victim model or a surrogate dataset to query the victim model and distill its knowledge; and *(b) Data-free Model Extraction (DFME)*, where no prior knowledge of the target victim model's training data is known and the synthesis of the attacker's query data is iteratively refined using the victim model's outputs as feedback. *(2) Architecture Stealing (Rolnick & Kording, 2020)*: the goal is to infer the internal design of the victim model, such as its layer structure or hyperparameters. Unlike functionality stealing, which focuses on prediction performance, architecture stealing targets the proprietary network design itself, enabling adversaries to reconstruct or optimize their own models.

## 5.2 Mathematical Formalization

**Attacker's Knowledge and Objective.** In a model extraction attack, the adversary interacts with a victim model $V(\boldsymbol{x}; \boldsymbol{\theta}_V)$ solely through its API. By submitting a set of queries $X = \{\boldsymbol{x}_i\}_{i=0}^{i=m}$, attacker receives corresponding outputs: $y_i = V(\boldsymbol{x}_i; \boldsymbol{\theta}_V), \quad i = 1, \ldots, m$, which may be either probability vectors (*soft-labels*) or top-1 predictions (*hard-labels*), depending on the API configuration. Using the collected pairs $\{(\boldsymbol{x}_i, y_i)\}_{i=1}^{i=m}$, the attacker then trains a substitute (clone) model $C(\boldsymbol{x}; \boldsymbol{\theta}_C)$ with the goal of reproducing the predictive behavior of $V$. Formally, the surrogate model's parameters are learned by solving: $\boldsymbol{\theta}_C^{\star} \in \arg\min_{\boldsymbol{\theta}_C} \sum_{i=1}^m \mathcal{L}(\boldsymbol{x}_i, y_i, \boldsymbol{\theta}_C)$, where $\mathcal{L}$ is an appropriate loss function (e.g., distillation loss or cross-entropy loss). Based on whether prior information of training data is available, MEAs can be categorized into *data-based* (DBME) and *data-free* (DFME) approaches.

**Defender's Knowledge and Objective** Defender's aim is to maintain the victim model's accuracy on its in-distribution (ID) dataset while simultaneously degrading the utility of any cloned model trained through the extraction attack. In practice, the defender operates under limited knowledge: the precise attack strategy, the architecture of the clone model, and whether a query is benign or adversarial are typically unknown. A common assumption is that adversarial queries originate from out-of-distribution (OOD) data (Kariyappa & Qureshi, 2020a), since the original training set is private and rarely exposed to API users. Nevertheless, effective defenses should also remain applicable when attackers have access to in-distribution queries, ensuring robustness across both DBME- and DFME-style attacks.

## 5.3 Taxonomy and Techniques of MEAs

We categorize model extraction attacks into two main types (*data-based* and *data-free*) which differ in whether the attacker has prior access to the victim model's training data.

### 5.3.1 Data-Based Attacks

Data-based extraction begins from public or domain-related inputs, leveraging semantic priors for faster convergence and higher sample utility.

*(a)* One stream of work focuses on query selection and augmentation. The seminal JBDA approach (Papernot et al., 2017) introduced Jacobian-based dataset augmentation, training surrogate model from black-box label outputs to approximate decision boundaries. *Copycat CNN* (Correia-Silva et al., 2018) shows that using non-problem-domain natural images can replicate the functionality of target models. *ActiveThief* (Pal et al., 2020) integrates K-center and active learning to select uncertain samples, achieving higher agreement with fewer queries. Augmentation and ensembles can further increase the informativeness of each query. For example, *Army of Thieves* (Jindal et al., 2024) employs ensemble-based consensus entropy and label disagreement to guide query selection. Meanwhile, *AugSteal* (Gao et al., 2024b) combines Grad-CAM-based data filtering and MPCL active selection with a FusionAug module (GridMix + MulAug) to enhance functional similarity under hard-label constraints. Finally, *MARICH* (Karmakar & Basu, 2024) matches the victim model output distribution via entropy- and divergence-based objectives, achieving high fidelity to the victim model's performance.

*(b)* A second stream explicitly probes the decision boundary of the victim model. *CloudLeak* (Yu et al., 2020) employs adversarial and active-learning–based queries to explore regions near classification boundaries, while *BEST* (Li et al., 2022) refines this idea with entropy-driven uncertain examples to capture both accuracy and robustness regions. *SPSG* (Zhao et al., 2024b) leverages superpixel segmentation and low-variance gradient estimation to approximate boundary information efficiently under limited queries. Other approaches, such as *InverseNet* (Gong et al., 2021) and *LOKT* (Nguyen et al., 2024a), reconstruct the victim's data distribution respectively through input inversion and label-space generative transfer.

*(c)* A third category of attacks leverages extra signals (e.g., explanations) to enhance the effectiveness of model extraction. Explanation-based attacks such as *XaMEA* (Yan et al., 2023) exploit explanation signals (e.g., saliency maps) to enhance surrogate fidelity, while *PtbStolen* (Zhang et al., 2023a) steals encoder representations via feature-vector matching on perturbed samples. At the system level, even when only hard labels are accessible, query-based knockoff and Jacobian-augmentation attacks remain feasible (Tramèr et al., 2016). Similarly, limited access to RNN hidden states can suffice for replication (Takemura et al., 2020).

**Foundation models (Data-Based)**    For LLMs, *in-context* imitation attacks (Li et al., 2024c) demonstrate that even without gradient access, medium-sized models can reproduce specialized abilities such as code summarization and synthesis by querying black-box APIs with carefully designed prompts. Here, traditional datasets are replaced by curated prompt sets and task suites, indicating that prompt programming itself can function as a data-based extraction strategy. Together with earlier observations on boundary probing and attribution-guided attacks, these results highlight that restricting access to the API interface alone is insufficient to prevent high-fidelity cloning.

### 5.3.2 Data-Free Attacks

Data-free model extraction (DFME) synthesizes $Q$ without access to in-distribution data. Then, the generation strategies accelerated progress: *MAZE* (Kariyappa et al., 2021a) guides a generator toward regions of maximal model disagreement via zeroth-order gradient estimation; and *DFMS-HL* (Sanyal et al., 2022) adapts to hard-label constraints with class-diversity regularization and adversarial alignment terms. *DFMS-DG* (He et al., 2024) utilizes denoising diffusion GANs to generate diverse, high-quality samples that improve clone model accuracy even against adversarially trained targets. DFCL-APIs (Yang et al., 2024a) extends the study of model extraction to the continual learning paradigm.

While generator-driven methods remain central to DFME, recent approaches enhance them with explicit optimization and sample selection strategies. For example, *ES Attack* (Yuan et al., 2022) and *Truong et al.* (Truong et al., 2021) iteratively refine synthetic queries through alternating estimation and synthesis steps to distill the victim model, while *DFHL-RS* (Yuan et al., 2024) generates high-entropy examples near decision boundaries and reuses them to reduce costs under hard-label constraints. Because DFME is highly budget-

sensitive, several methods emphasize query efficiency: *IDEAL* (Zhang et al., 2022) decouples generation from distillation, requiring only one query per synthetic sample, whereas $E^3$ (Zhu et al., 2025a) improves efficiency via language-guided sample selection, multi-resolution training, and temperature tuning, achieving comparable fidelity with merely about 0.5% of the queries needed by conventional GAN-based methods; and *DualCOS* (Yang et al., 2024b) incorporates active sampling and disagreement-based objectives to maximize sample reuse. Under label-only APIs, *QUDA* (Lin et al., 2023) employs deep reinforcement learning with weak generative priors; and *DisGUIDE* (Rosenthal et al., 2023) drives divergence in clone outputs to increase informativeness while reducing query counts. DFME also extends beyond classification: *Yue et al.* (Yue et al., 2021) demonstrate that generating synthetic queries can effectively replicate victim recommender models.

**Foundation models (Data-Free)** DFME is no longer vision-only. In language, *Lion* (Jiang et al., 2023b) employs adversarial knowledge distillation with a feedback loop that generates hard instructions to efficiently distill GPT-style models using only 70K queries, achieving ChatGPT-level performance with minimal supervision. Random or task-agnostic queries can also suffice for BERT-based APIs (Krishna et al., 2020). Beyond text, *LCA* (Shao et al., 2025a) leverages Stable Diffusion's latent prior for adversarial query synthesis, combining latent augmentation with membership-aware sampling to produce high-utility prompts in text and multimodal settings.

In both data-based and data-free settings, recent studies (Jagielski et al., 2020; He et al., 2021; Dai et al., 2023) have proposed extraction methods that generalize well across domains and remain effective even with a limited number of queries. Foundation models further amplify these risks because their prompt interaction interfaces and few-shot generalization make high-fidelity cloning more practical. These developments highlight the need for defense-in-depth strategies—including dynamic output perturbation, query-pattern monitoring, attribution filtering, and robust watermarking—that are adapted to the attacker's query regime.

## 5.4 Defensive Techniques.

Existing model extraction defenses can be broadly categorized into two classes: *prevention defenses*, which actively degrade the extraction process, and *verification/detection defenses*, which passively monitor or verify whether model extraction has occurred.

### 5.4.1 Model extraction prevention defenses (active defenses)

These defenses aim to reduce the value of queries or limit the fidelity of stolen models. Representative strategies include:

*(a) Model output perturbation.* A primary line of work modifies API outputs to reduce the information available to an attacker. Techniques include selective output perturbation, response filtering, or adaptive shaping that perturbs outputs only for abnormal queries while preserving accuracy on legitimate ones. Examples include *AdvFT* (Zhang et al., 2024), which perturbs feature representations of out-of-distribution (OOD) queries; *CIP* (Zhang et al., 2023b), which combines energy-based OOD scoring with selective poisoning and traceable watermarking; *AMAO* (Jiang et al., 2023a), which integrates adversarial training, adaptive outputs, and embedded watermarks; *ModelGuard* (Tang et al., 2024), which adaptively optimizes perturbations to balance information leakage and prediction utility; and *Noise Transition Matrices* (Wu et al., 2024), which inject lightweight structured noise.

*(b) Training-time defenses.* Some defenses alter the model itself during training to make model extraction harder. Defensive training strategies (Wang et al., 2023b; Hong et al., 2024) inject robustness by learning causal or distributionally robust representations that hinder surrogate learning. Architectural modifications include *DNF* (Luan et al.) with early exits, and *InI* (Guo et al., 2023) which dynamically isolates suspicious queries to block gradient-based extraction. *MeCo* (Wang et al., 2023b) further employs distributionally robust defensive training with a data-dependent perturbation generator to resist data-free extraction. AAUG (Wang et al., 2024a) which introduces an attack-aware and uncertainty-guided training objective to reduce the accuracy of stolen models, and Beowulf (Gong et al., 2024) introduces adversarial dummy classes during training, which reshape decision boundaries to mislead surrogate models and degrade the fidelity of surrogate models. RL-based meta-policies (Orekondy et al., 2019) adapt label shaping dynamically, while

ensemble defenses (Kariyappa et al., 2021b) diversify decision boundaries to resist approximation. Active watermarking (Wang et al., 2024b) fine-tunes models to embed probabilistic signals that actively degrade the performance of cloned models.

### 5.4.2 Model extraction verification/detection defenses (passive defenses)

These defenses do not prevent model extraction directly, but provide evidence or detection signals.

*(a) Query-level monitoring and anomaly detection.* Detection-based methods identify extraction attempts in real time. Statistical defenses such as *PRADA* (Juuti et al., 2019) analyze query distributions, while *SEAT* (Zhang et al., 2021) and *HODA* (Sadeghzadeh et al., 2023) leverage query similarity to detect structured exploration. More advanced anomaly detectors include *SAME* (Xie et al., 2024), which uses autoencoder reconstruction error, and adaptive detection-and-response methods (Kariyappa & Qureshi, 2020a). Although effective, these methods can be bypassed if adversaries disguise their query distribution (Azizmalayeri et al., 2022).

*(b) Verification-based watermarking.* Another class of defenses embeds verifiable signals into models so that ownership can be established after suspected theft. Earlier trigger-set or boundary-based approaches (Adi et al., 2018; Zhang et al., 2018) and subsequent schemes such as *DeepSigns* (Darvish Rouhani et al., 2019) enable IP owners to query a suspect model and check for expected responses. These methods include both training-time and API-level watermarking, providing strong ownership verification but offering no direct prevention against model extraction. For generative models, *WDM* (Peng et al., 2023) embeds watermarks into diffusion U-Nets, while adversarial or lexical watermarking (He et al., 2022) inserts detectable linguistic or statistical patterns that later facilitate ownership verification. Representative approaches such as *EWE* (Jia et al., 2021) and *DAWN* (Szyller et al., 2021) further embed persistent signals that survive in surrogate models, thereby enabling reliable ownership tracing. More recently, *LIDet* (Zhao et al.) extends this line of work to large language models, using text-level watermarks to detect intellectual property infringement in suspect LLMs under black-box access.

In summary, model extraction defenses fall into two complementary categories. *Prevention defenses* actively interfere with the attacker by shaping API outputs, or modifying training objectives, thereby lowering the fidelity or usability of stolen models. *Verification and detection defenses*, in contrast, do not stop extraction but provide monitoring signals (e.g., anomaly detection on queries) or verifiable marks for post-hoc attribution. In practice, robust protection often combines both: prevention raises the attack cost and reduces the value of extraction, while verification provides the evidence and legal traceability once theft has occurred.

## 5.5 From Functional Imitation to Structural Theft

So far, Model→Model ($M \rightarrow M$) discussion has focused on model *functional* cloning, where the goal is to reproduce the victim's input–output behavior. More recent attacks target *structural* recovery, inferring model architectures, parameters, or training settings. This shift changes the objective (from output agreement to structural fidelity) and increases risk: recovering model internals lowers the cost of subsequent attacks and can defeat defenses that rely solely on restricting API outputs.

**Structural and Parametric Extraction** *(a) From structural inference to architecture reconstruction.* Early studies demonstrated that black-box outputs can leak model architecture and training attributes. Rolnick et al. (Rolnick & Kording, 2020) exploited the piecewise linearity of ReLU networks to analytically reconstruct their layer topology and parameters, yielding functionally equivalent models. *(b) Practical weight recovery via learning and analysis.* Jagielski et al. (Jagielski et al., 2020) combined surrogate imitation with direct analytical recovery, achieving near-exact weight reconstruction for multi-layer ReLU networks using only a few thousand queries. Carlini et al. (Carlini et al., 2020) reformulated parameter recovery as a differential-cryptanalysis problem, highlighting the feasibility of fine-grained extraction under black-box access. *(c) Partial reconstruction in large-scale production models.* Extending to commercial settings, Carlini et al. (Carlini et al., 2024b) showed that even limited API feedback (e.g., log probabilities or logit bias) suffices

Table 2: **Comparison of attack categories in the unified closed-loop taxonomy.** We summarize threat-model assumptions, attacker capabilities, typical evaluation metrics, and computation cost across the four directional attack classes (D→D, D→M, M→D, M→M).

| Attack class | Threat-model assumptions | Attacker capabilities required | Typical success metrics | Computational cost |
|---|---|---|---|---|
| **D→D** | Access to *protected* data or prompts; goal is to bypass ownership or safety constraints while preserving content or utility. | Ability to apply transformations to inputs (regeneration, rewriting, adversarial prompting, etc); usually no need for victim model weights. | Bypass success rate (e.g., decrypt success; watermark detection failure; jailbreak success), plus fidelity or utility preservation and downstream task performance. | Low to moderate computation cost: input transformation or rewriting; jailbreak often needs iterative prompt search; watermark removal may involve generative regeneration. |
| **D→M** | Adversary can influence training or fine-tuning data to implant targeted misbehavior or erode alignment. | Inject or modify a subset of training samples to create malicious training data. | Attack success (attack success rate, targeted misclassification, backdoor trigger success); clean utility (clean accuracy, helpfulness); safety metrics (refusal rate drop, unsafe prompt completion rate). | Moderate to high computation cost: requires training or fine-tuning data modification by optimization. |
| **M→D** | Adversary queries a trained model to recover memorized training information (samples, attributes or membership). | Black-box (query outputs) or white-box (gradients or representations) access; optimization over inputs or prompts. | Data reconstruction quality; membership information accuracy/AUC. | Often high computation cost: many queries and/or iterative optimization; generative extraction can require generative model training. |
| **M→M** | Adversary aims to replicate a deployed proprietary model via queries (data-free/data-based/architecture cloning). | Query access to victim; ability to train surrogate; optionally synthesize queries through optimization or generative models; sometimes partial architecture knowledge. | Fidelity and utility of stolen model (accuracy, KL divergence of logit matching); query efficiency (e.g., number of queries). | High computation cost: dominated by query budget and surrogate training compute; adaptive query synthesis can reduce queries but adds generation cost. |

to reconstruct key components of large-scale LLMs with high fidelity and modest cost, underscoring practical risks for deployed models.

# 6 Attacks and Defense Comparisons, Interactions and Inter-dependencies

## 6.1 Attacks and Defenses Comparisons

We map representative defense strategies to the attack classes they are capable of mitigating in Table 3. Rather than treating defenses in isolation, this table highlights how different defense families, including data sanitization, defensive training, watermarking, output perturbation and alignment fine-tuning—interact with multiple attacks. This mapping therefore offers practitioners a practical guide for understanding which defenses apply to which threat classes. It reveals that most defenses are not isolated countermeasures, but instead influence multiple attacks.

Table 3: Mapping of defense methods to the attacks they address, with effectiveness and limitations.

| Defense Method | Attack(s) Addressed | Effectiveness | Limitations |
|---|---|---|---|
| Data Sanitization | data poisoning attack, backdoor attack, harmful fine-tuning attack | Partially effective | Ineffective once a model is trained; vulnerable to adaptive or clean-label poisoning; may remove useful data and hurt utility |
| Defensive training | Adversarial attack, data poisoning attack, model inversion attack, model extraction attack and membership inference attack | Partially effective | High computational cost; limited scalability; may reduce accuracy on clean data |
| Watermarking | model extraction attack (ownership verification) | Detection-only (passive) | passive and cannot prevent model extraction from happening |
| Output perturbation | model extraction attack, membership inference attack | Partially effective | Degrades utility; vulnerable to adaptive attacks; costly at inference |
| Alignment fine-tuning | harmful fine-tuning attack, jailbreak attack | Partially effective | Vulnerable to prompt-based bypasses and adversarial fine-tuning; expensive and not robust under distribution shift |
| Detection | Adversarial attack, harmful fine-tuning attack, jailbreak attack, model extraction attack | not preventive | Reactive only; requires reliable detection signals |

## 6.2 Attack Interactions and Dependencies

A key insight of our unified closed-loop perspective is that security threats in foundation models do not operate in isolation. Instead, attacks across the data space (D) and model space (M) are *interdependent*, forming feedback loops in which vulnerabilities in one class systematically facilitate, amplify or weaken threats in another. We identify several cross-class connections that substantiate this closed-loop unified view.

**Poisoning (D→M) → Membership Inference (M→D)** Data poisoning attacks manipulate the training or fine-tuning data to corrupt the learned feature–label relationships encoded in model parameters (D→M). Beyond degrading predictive performance, such corruption can amplify privacy attacks. In particular, poisoned representations can increase overfitting, making it easier for adversaries to exploit output statistics for membership inference (M→D) (Chen et al., 2022; Wen et al., 2024b). The resulting privacy leakage further illustrates how model manipulation in the training phase can manifest as data exposure at inference time.

**Model Extraction (M→M) → Membership Inversion, Training Data Recovery (M→D) and Adversarial Data Generation (D→D).** Successful model extraction yields a high-fidelity surrogate model that approximates the behavior of the target system (M→M). This surrogate not only enables training data recovery attacks and model inversion (M→D) (Tramèr et al., 2016), but can also be used to synthesize adversarial data samples that closely follow the original training data distribution (D→D) (Papernot et al.,

2017). Such adversarially generated data may subsequently bypass data filtering or safety checks, illustrating the interaction between model compromise and data-space exploitation.

**Model Inversion (M→D) → Watermark Removal (D→D).** Model inversion attacks aim to reconstruct training data or data-like samples from a deployed model. Although the recovered samples may not be exact replicas of the original training instances, they often preserve salient semantic and distributional characteristics of the underlying data. As a result, an adversary can retrain a new model using inversion-generated data and claim that the model was trained solely on independently obtained, *legitimate* data since it is not directly traceable to the original training set. This effectively enables watermark removal and ownership obfuscation, as the resulting model may evade existing IP verification or data provenance mechanisms (Zong et al., 2024).

**Backdoor Attacks (D→M) weakens Model Extraction (M→M)** (Wang et al., 2025) proposes a defensive strategy that intentionally poisons the model extraction process by embedding a backdoor into any surrogate model trained from stolen outputs. Instead of preventing queries, the defender modifies the victim model's output so that its predictions subtly encode malicious signals while preserving normal task accuracy for benign users. When an attacker performs model extraction using these outputs, the resulting surrogate unintentionally learns a hidden backdoor, which can later be triggered to induce targeted misbehavior. This effectively makes model extraction vulnerable to controlled misuse and demonstrates that backdoor mechanisms can serve as an active defense against model extraction by poisoning the downstream learning process.

**Harmful Fine-Tuning (D→M) → Jailbreak Attack (D→M).** Harmful fine-tuning attacks directly modify model parameters by injecting malicious or misaligned data during fine-tuning (D→M). Such parameter-space corruption often weakens alignment and safety constraints. As a result, the fine-tuned model becomes more susceptible to jailbreak-style prompt manipulations in deployment, enabling harmful outputs Qi et al. (2024). In this case, a model-space attack reshapes the data-space vulnerability surface, feeding back into D→D misuse.

## 7 Other Categorization Perspectives

### 7.1 Attack Objectives: Privacy, Integrity, and Availability

From the perspective of security consequences, AI security attacks are commonly categorized into three classes: privacy, integrity, and availability attacks. *Privacy attacks* aim to extract or infer sensitive information about the training data, model parameters, or data subjects without authorization. In machine learning, this includes attacks such as model inversion, membership inference, and model extraction, which compromise confidentiality even when the system functions normally. *Integrity attacks* seek to manipulate or corrupt the model's behavior or outputs, causing it to produce incorrect, biased, or attacker-controlled predictions. Examples include data poisoning, backdoor attacks, and adversarial examples, which undermine the correctness and trustworthiness of the model. *Availability attacks* aim to degrade or deny access to the model or service, making it unusable or unreliable for legitimate users. In ML systems, this includes denial-of-service (DoS) attacks, resource-exhaustion via crafted inputs, etc.

While our taxonomy organizes attacks based on their interaction with the data and model components, the classical security objectives (privacy, integrity and availability) provide an orthogonal lens centered on attacker goals. This dimension focuses on the intended consequence of an attack. Taken together, the combination of our structural taxonomy and the these objectives provides a multi-dimensional understanding of the modern AI threat landscape. We provide this categorization and comparison in Table 4.

### 7.2 Attack Timing: Training-Time, Inference-Time, and Deployment-Time Attacks

Attack timing provides another orthogonal dimension that describes when adversaries interact with the system. This perspective complements our primary taxonomy by clarifying the lifecycle of an attack.

Table 4: Mapping representative attacks in our taxonomy to security objectives (privacy, integrity, availability). A checkmark indicates that the attack commonly targets the corresponding security objective.

| Attack | Privacy | Integrity | Availability |
|---|---|---|---|
| Adversarial examples | ✗ | ✓ | ✗ |
| Data decryption attack | ✓ | ✗ | ✗ |
| Watermark removal attack | ✗ | ✓ | ✗ |
| Data poisoning | ✗ | ✓ | ✗ |
| Harmful fine-tuning | ✗ | ✓ | ✗ |
| Jailbreak | ✗ | ✓ | ✗ |
| Training data extraction | ✓ | ✗ | ✗ |
| Model extraction | ✓ | ✗ | ✗ |
| Model inversion | ✓ | ✗ | ✗ |
| Membership inference | ✓ | ✗ | ✗ |

Table 5: Mapping representative attack categories in our taxonomy to attack timing (training / inference / deployment). A checkmark indicates that the attack can be instantiated at the corresponding phase.

| Attack Category | Training-time | Inference-time | Deployment-time |
|---|---|---|---|
| Adversarial examples | ✗ | ✓ | ✗ |
| Data decryption | ✓ | ✗ | ✗ |
| Watermark Removal | ✓ | ✓ | ✗ |
| Data poisoning | ✓ | ✗ | ✗ |
| Jailbreak | ✗ | ✓ | ✓ |
| Harmful fine-tuning | ✓ | ✗ | ✗ |
| Training data extraction | ✗ | ✓ | ✓ |
| Model extraction | ✗ | ✓ | ✓ |
| Model inversion | ✗ | ✓ | ✓ |
| Membership inference | ✗ | ✓ | ✓ |

- *Training-time attacks* modify the data or optimization process during training (e.g., poisoning, backdoors, harmful fine-tuning).

- *Inference-time attacks* exploit crafted queries to manipulate predictions or extract information during inference (e.g., adversarial examples, extraction, jailbreak prompting).

- *Deployment-time attacks* target system updates, configurations, and integration points (e.g., model extraction through API queries, membership inference).

We provide this categorization and comparison in Table 5.

# 8 Discussion and Future Directions

This survey advocates a *closed-loop* view of AI security in the era of foundation models, where data- and model-centric attacks are deeply interconnected rather than isolated. While existing studies have made important progress on individual threats, real-world deployments increasingly face *simultaneous, adaptive, and composed attacks.* Below, we outline concrete and open research directions that move beyond single-attack analysis and toward holistic, system-level defenses.

### 8.1 Defending Against Simultaneous and Composed Attacks

*Open Problems.* Most existing defenses are designed and evaluated against a *single attack* (e.g., extraction *or* jailbreak *or* inversion). A critical open problem is how to design defenses that are *robust to multiple attacks occurring concurrently or sequentially.* Several important open questions remain unresolved.

- How can a defense mitigate multiple attack objectives simultaneously without assuming prior knowledge of which attack is active?

- How should defenses adapt when attackers jointly optimize multiple goals, such as maintaining jailbreak success while extracting model information?

*Technical Barriers.* Defending against multiple attacks raises several fundamental challenges:

- *Multiple defense interference*: Defenses optimized for one threat can degrade robustness against others (e.g., adversarial training may inadvertently amplify the effectiveness of model extraction attacks. (Khaled et al., 2022)). This occurs because many attacks share underlying internal representations, so mitigating one vulnerability can unintentionally weaken protection against another.

- *Conflicting objectives:* Security, alignment, privacy, and utility objectives are often incompatible, making joint optimization difficult.

*Evaluation Methodology.* We argue that future work should adopt *multi-attack testing* protocols, where:

- Multiple attacks are launched concurrently or sequentially (e.g., jailbreak + model extraction, data poisoning → membership inference).

- Defenses are evaluated using joint metrics such as worst-case attack success, cross-attack amplification, and utility degradation.

- At least one defense-aware adaptive multi-attack attacker is included to test robustness under strategic adaptation.

### 8.2 Closed-Loop Defense Design and Adaptation

*Open Problems.* A promising direction is to move from static defenses to *closed-loop adaptive defenses* that evolve with model updates and attacker behavior. Open problems include: (1) Online detection of attacker behavior drift under non-stationary query distributions. (2) Learning defense policies that dynamically choose interventions (e.g., refusal, perturbation, detection) under utility and latency constraints. (3) Continual recalibration of defenses after fine-tuning, alignment updates, or retrieval-augmented deployment.

*Technical Barriers.* Several challenges include delayed feedback about attack success, high false-positive rates under attack data distribution shifts, and the risk that adaptive defenses introduce new vulnerabilities.

*Evaluation Methodology.* We recommend streaming evaluation protocols that measure detection success rate, cumulative harm, and long-term utility under adaptive attackers rather than static test sets.

### 8.3 Joint Optimization of Security, Alignment, and Utility

*Open Problems.* Most defenses implicitly optimize a single axis (e.g., privacy or alignment). An important open problem is how to *jointly optimize* security, alignment, and utility. We expect to formalize multi-objective optimization frameworks that expose trade-offs explicitly and develop harm-aware metrics that go beyond attack success rate to measure downstream impact.

*Technical Barriers.* The lack of composable guarantees across different defense mechanisms and the mismatch between existing metrics and real-world harm remain major obstacles.

*Evaluation Methodology.* Future evaluations should report full trade-off curves, including utility loss, compute overhead, robustness under adaptive attacks, failure severity and alignment.

Table 6: Comparison of different Watermark Removal attack methods on LVW and CLWD datasets.

| Dataset | Methods | PSNR | SSIM | RMSE | $\mathrm{RMSE}_w$ |
|---------|---------|------|------|------|-------|
| LVW | U-Net (Ronneberger et al., 2015) | 30.33 | 0.9517 | 7.11 | 42.18 |
| | Attentive recurrent network (Qian et al., 2018) | 39.92 | 0.9902 | 3.31 | 21.40 |
| | DHAN (Cun et al., 2020) | 40.68 | 0.9949 | 2.62 | 17.29 |
| | cGAN-based (Li et al., 2019) | 33.57 | 0.9690 | 5.84 | 34.71 |
| | VWGAN (Cao et al., 2019) | 34.16 | 0.9714 | 5.51 | 33.42 |
| | WDNet (Liu et al., 2021) | 42.45 | 0.9954 | 2.39 | 12.75 |
| | BVMR (Hertz et al., 2019) | 40.14 | 0.9910 | 3.24 | 18.57 |
| | SplitNet (Cun & Pun, 2021) | 43.16 | 0.9946 | 2.28 | 14.06 |
| | SLBR (Liang et al., 2021) | 43.48 | 0.9959 | 2.15 | 12.14 |
| CLWD | U-Net (Ronneberger et al., 2015) | 23.21 | 0.8567 | 19.35 | 48.43 |
| | Attentive recurrent network (Qian et al., 2018) | 34.60 | 0.9694 | 5.40 | 19.34 |
| | DHAN (Cun et al., 2020) | 35.29 | 0.9712 | 5.28 | 18.25 |
| | cGAN-based (Li et al., 2019) | 27.96 | 0.9161 | 12.63 | 46.80 |
| | VWGAN (Cao et al., 2019) | 29.04 | 0.9363 | 10.36 | 41.21 |
| | WDNet (Liu et al., 2021) | 35.53 | 0.9738 | 5.11 | 17.27 |
| | BVMR (Hertz et al., 2019) | 35.89 | 0.9734 | 5.02 | 18.71 |
| | SplitNet (Cun & Pun, 2021) | 37.41 | 0.9787 | 4.23 | 15.25 |
| | SLBR (Liang et al., 2021) | 38.28 | 0.9814 | 3.76 | 14.07 |
| | DVW (Zhang et al., 2025a) | 28.52 | 0.9445 | 11.74 | 46.29 |
| | IMPRINTS (Chen et al., 2025b) | 50.61 | 0.9993 | 0.96 | 3.56 |

## 8.4 Benchmarking

A major gap is the absence of unified benchmarks that reflect closed-loop and multi-attack realities. We expect to build comprehensive benchmarks that explicitly model attack chaining across data and model stages. We also expect standardized reporting of budgets, access levels, and defense costs.

*Outlook.* Developing shared benchmarks and evaluation protocols for simultaneous and composed attacks will be essential for advancing trustworthy foundation models. We hope this survey motivates the community to move beyond isolated threat analysis toward holistic, closed-loop security research.

# 9 Experimental Evaluation

## 9.1 Empirical Evaluation for Various Attacks

**Empirical Evaluation for D→D**    Table 6 presents a quantitative comparison of representative watermark removal methods on the LVW and CLWD datasets, which differ in watermark complexity from grayscale patterns to diverse colored images. The evaluated approaches include conditional GAN-based methods, attention-based models, and general image content removal networks adapted for watermark removal, and are evaluated using PSNR, SSIM, RMSE, and weighted RMSE under a unified experimental protocol, following the evaluation setup in (Liang et al., 2021).

**Empirical Evaluation for D→M**    Table 7 evaluates model robustness under different harmful data ratios in the user fine-tuning stage, following the harmful fine-tuning threat model in (Huang et al., 2024c). Specifically, models are first aligned using supervised fine-tuning on safe samples from the BeaverTails dataset, and then fine-tuned on downstream benign tasks where a proportion $p$ of harmful instructions from BeaverTails is injected into the training data. Experiments are conducted with a fixed fine-tuning sample size ($n = 1000$ by default), and performance is measured using Harmful Score and Fine-tune Accuracy on the corresponding downstream tasks, with results averaged across multiple independent runs.

Table 7: Performance under different harmful ratio.

| Methods | Harmful Score ↓ | | | | | | Fine-tune Accuracy ↑ | | | | | |
|---|---|---|---|---|---|---|---|---|---|---|---|---|
| (n=1000) | clean | p=0.01 | p=0.05 | p=0.1 | p=0.2 | Avg. | clean | p=0.01 | p=0.05 | p=0.1 | p=0.2 | Avg. |
| Non-Aligned | 34.20 | 65.60 | 81.00 | 77.60 | 79.20 | 67.52 | 95.60 | 94.60 | 94.00 | 94.60 | 94.40 | 94.64 |
| SFT (Li et al., 2023) | 48.60 | 49.80 | 52.60 | 55.20 | 60.00 | 53.24 | 94.20 | 94.40 | 94.80 | 94.40 | 94.20 | 94.40 |
| EWC (Kirkpatrick et al., 2017) | 50.60 | 50.60 | 50.60 | 50.60 | 50.60 | 50.60 | 88.60 | 88.20 | 87.40 | 86.80 | 80.60 | 86.32 |
| VIguard (Zong et al.) | 49.40 | 50.00 | 54.00 | 54.40 | 53.60 | 60.20 | 94.80 | 94.80 | 94.60 | 94.60 | 94.60 | 94.68 |
| KL (Huang et al., 2024c) | 54.40 | 53.60 | 55.20 | 54.00 | 56.60 | 54.76 | 85.80 | 85.80 | 85.00 | 85.40 | 84.60 | 59.08 |
| Vaccine (Huang et al., 2024c) | 42.40 | 42.20 | 42.80 | 48.20 | 56.60 | 46.44 | 92.60 | 92.60 | 93.00 | 93.80 | 95.00 | 93.40 |
| Lisa (Huang et al., 2024a) | 53.00 | 60.90 | 64.80 | 68.20 | 72.10 | 63.80 | 93.92 | 93.69 | 93.58 | 93.23 | 91.17 | 93.12 |
| Repnoise (Rosati et al., 2024a) | 66.50 | 77.60 | 78.80 | 78.60 | 77.90 | 75.88 | 89.45 | 92.66 | 93.69 | 94.72 | 94.38 | 92.98 |
| LDIFS (Mukhoti et al.) | 51.70 | 67.70 | 68.80 | 72.30 | 71.80 | 66.46 | 93.46 | 93.23 | 93.69 | 93.23 | 94.04 | 93.53 |
| Antidote (Huang et al., 2025b) | 52.90 | 61.20 | 61.20 | 64.60 | 64.50 | 60.88 | 93.58 | 93.46 | 93.12 | 93.35 | 91.74 | 93.05 |

Table 8: The overall results on the role-play and safety benchmarks with LLaMA-3-8B-Instruct, and Gemma-2-9b-it under the LoRA fine-tuning settings. The results are the average performance across 10 roles.

| LoRA Fine-tuning | RoleBench ↑ | | | Safety ↑ | | | | Jailbreak ↑ |
|---|---|---|---|---|---|---|---|---|
| | RAW | SPE | AVG. | AdvBench | BeaverTails | HEx-PHI | AVG. | AVG. |
| **LLaMA-3-8B-Instruct** | | | | | | | | |
| LLaMA-3-8B-Instruct baseline | 23.86 | 19.14 | 21.50 | 98.46 | 91.40 | 95.33 | 95.06 | 78.80 |
| SFT (Ouyang et al., 2022) | 30.42 | 24.82 | 27.62 | 87.75 | 78.77 | 83.43 | 83.32 | 73.26 |
| SPPFT (Li et al., 2025b) | 29.48 | 24.49 | 26.99 | 86.23 | 78.95 | 81.47 | 82.22 | 58.18 |
| SafeLoRA (Hsu et al., 2024) | 29.64 | 23.84 | 26.74 | 88.75 | 80.88 | 85.17 | 84.93 | 72.28 |
| SafeInstr (Bianchi et al., 2024) | 30.14 | 24.55 | 27.35 | 92.85 | 79.41 | 87.67 | 86.64 | 73.00 |
| Vaccine (Huang et al., 2024c) | 26.16 | 20.05 | 20.89 | 96.37 | 87.43 | 94.63 | 83.36 | 79.80 |
| SEAL (Shen et al., 2025) | 26.32 | 22.09 | 24.21 | 97.54 | 90.53 | 93.20 | 93.76 | 64.82 |
| SaRFT (Zhao et al., 2025) | 29.35 | 23.41 | 26.38 | 97.62 | 89.56 | 93.47 | 93.55 | 80.08 |
| **Gemma-2-9b-it** | | | | | | | | |
| Gemma-2-9b-it baseline | 23.37 | 16.44 | 19.91 | 99.42 | 95.20 | 99.67 | 98.10 | 30.40 |
| SFT (Ouyang et al., 2022) | 31.69 | 27.19 | 29.44 | 85.04 | 85.79 | 86.27 | 85.69 | 25.96 |
| SPPFT (Li et al., 2025b) | 31.63 | 27.20 | 29.42 | 85.94 | 86.24 | 86.60 | 86.26 | 26.74 |
| SafeLoRA (Hsu et al., 2024) | 32.47 | 26.62 | 29.54 | 87.67 | 89.75 | 89.57 | 88.99 | 27.44 |
| SafeInstr (Bianchi et al., 2024) | 32.02 | 26.95 | 29.49 | 87.40 | 82.08 | 85.13 | 84.87 | 26.02 |
| Vaccine (Huang et al., 2024c) | 21.28 | 17.92 | 19.60 | 90.12 | 91.74 | 92.27 | 91.38 | 22.46 |
| SEAL (Shen et al., 2025) | 29.06 | 23.43 | 26.25 | 98.08 | 97.00 | 96.77 | 97.28 | 25.12 |
| SaRFT (Zhao et al., 2025) | 30.78 | 25.23 | 28.01 | 98.27 | 95.58 | 97.10 | 96.98 | 30.70 |

Complementarily, Table 8 reports the overall role-playing, safety, and jailbreak robustness under LoRA fine-tuning, following the experimental setup in (Zhao et al., 2025). Evaluations are performed on two instruction-tuned backbones, namely LLaMA-3-8B-Instruct and Gemma-2-9B-it, under a unified LoRA configuration. Role-play performance is measured on RoleBench using RAW and SPE scores, while safety is assessed on AdvBench, BeaverTails, and HEx-PHI, together with jailbreak robustness evaluated across multiple attack settings. All results are averaged over 10 representative roles.

**Empirical Evaluation for M→D** Table 9 reports the performance of representative membership inference attack (MIA) methods on the WIKIMIA benchmark across multiple pre-trained LLMs, following the experimental setup in (Fu et al., 2025). The evaluation covers seven target LLMs spanning different model families. All methods are evaluated using AUC as the primary metric. Overall, the results highlight the varying effectiveness of existing MIA techniques under different attack paradigms and model architectures.

**Empirical Evaluation for M→M** Table 10 summarizes the teacher and student classification accuracies of representative model extraction attacks under the M→M setting on the MNIST and CIFAR-10 datasets,

Table 9: Membership inference attack (MIA) comparison between methods for LLMs on WIKIMIA dataset, "—" indicates that the data are unavailable.

| Method | Pythia | Falcon | LLaMA-2 |
|---|---|---|---|
| PPL (Yeom et al., 2018) | 0.69 | 0.62 | 0.61 |
| Min-K% (Shi et al.) | 0.74 | 0.64 | 0.64 |
| Min-K%++ (Zhang et al., 2025b) | 0.66 | 0.74 | 0.57 |
| Zlib (Carlini et al., 2021) | 0.72 | 0.64 | 0.63 |
| Lowercase (Carlini et al., 2021) | 0.69 | 0.63 | 0.61 |
| Neighbor (Mattern et al., 2023) | 0.66 | 0.59 | 0.61 |
| MIA-Tuner (Fu et al., 2025) | 0.96 | 0.91 | 0.98 |
| Random Swapping (RS) attack (Tang et al., 2023) | 0.50 | 0.51 | 0.50 |
| Back Translation (BT) attack (Tang et al., 2023) | 0.51 | 0.48 | 0.49 |
| Word Substitution (WS) attack (Tang et al., 2023) | 0.49 | 0.51 | 0.51 |
| SMIA (Mozaffari & Marathe) | 0.67 | — | — |
| BEAST (Sadasivan et al., 2024) | 0.63 | — | 0.55 |
| SaMIA (Kaneko et al., 2025) | — | — | 0.7 |
| PETAL (He et al., 2025) | 0.64 | 0.60 | 0.58 |

Table 10: MNIST and CIFAR-10 ACC Soft-Label

| Dataset | Methods | T Architecture | Teacher | S Architecture | Student |
|---|---|---|---|---|---|
| MNIST | GAME (Xie et al., 2022) | LeNet | 98.74 | half-LeNet | 90.36 |
| MNIST | ES Attack (Yuan et al., 2022) | LeNet5 | 92.03 | ResNet18 | 98.13 |
| MNIST | IDEAL (Zhang et al., 2022) | LeNet | 99.27 | LeNet | 96.32 |
| MNIST | TandemGAN (Hong et al., 2023) | VGG16 | 99.51 | VGG11 | 91.30 |
| CIFAR-10 | TandemGAN (Hong et al., 2023) | ResNet34 | 90.71 | VGG11 | 29.58 |
| CIFAR-10 | QUDA (Lin et al., 2023) | ResNet18 | 81.74 | ResNet18 | 68.99 |
| CIFAR-10 | MAZE (Kariyappa et al., 2021a) | ResNet20 | 92.26 | WideResNet | 89.85 |
| CIFAR-10 | DFMS-HL (Sanyal et al., 2022) | ResNet34 | 95.59 | ResNet18 | 91.24 |
| CIFAR-10 | ES Attack (Yuan et al., 2022) | ResNet34 | 91.93 | ResNet18 | 62.73 |
| CIFAR-10 | IDEAL (Zhang et al., 2022) | ResNet34 | 93.85 | ResNet34 | 68.82 |
| CIFAR-10 | E3 (Zhu et al., 2025a) | ResNet34 | 95.94 | ResNet18 | 94.01 |
| CIFAR-10 | DualCOS (Yang et al., 2024b) | ResNet34 | 95.54 | ResNet18 | 94.86 |
| CIFAR-10 | MEGEX (Miura et al., 2024) | ResNet34 | 95.54 | ResNet18 | 91.61 |

following the experimental setup in DFMS-HL (Sanyal et al., 2022). Across both datasets, teacher models consistently maintain high accuracy, while the student performance exhibits substantial variation depending on the attack strategy and the architectural relationship between teacher and student models. On CIFAR-10, recent extraction methods such as E3 (Zhu et al., 2025a) and DualCOS (Yang et al., 2024b) achieve student accuracies that closely approach their corresponding teachers, whereas earlier approaches—particularly those involving heterogeneous architectures—often suffer from notable performance degradation.

Table 11 reports the defensive performance of representative defenses against a diverse set of model extraction attacks on a CIFAR-10 target model, following the experimental protocol of ModelGuard (Tang et al., 2024). The evaluation considers two query strategies (KnockoffNet (Orekondy et al., 2019) and JBDA-TR (Juuti et al., 2019)), and measures defense effectiveness by the extraction accuracy of the substitute model under fixed utility constraints. This unified setup enables a fair comparison between classical defenses (None, RevSig (Lee et al., 2019), MAD (Orekondy et al., 2020), AM (Kariyappa & Qureshi, 2020b), Top-1 (Sanyal et al., 2022), Rounding (Tang et al., 2024), recent defense work such as MGW (ModelGuard-W), MGS (ModelGuard-S) (Tang et al., 2024), and QUEEN method (Chen et al., 2025a).

## 9.2 Case Study

To illustrate AI security in practice, we construct a medical image classification pipeline using the PathM-NIST in MedMNIST dataset (Yang et al., 2021; 2023a), a large-scale and standardized benchmark widely

Table 11: Defensive performance of different defense methods against different attacks on the target model trained on CIFAR-10.

| Query Method | Attack Method | None | RevSig | MAD | AM | Top-1 | Rounding | MGW | MGS | QUEEN |
|---|---|---|---|---|---|---|---|---|---|---|
| KnockoffNet | Naive Attack (Truong et al., 2021) | 87.32% | 84.86% | 83.61% | 82.84% | 83.51% | 86.91% | 75.06% | 84.11% | 10.00% |
| | Top-1 Attack (Sanyal et al., 2022) | 83.51% | 83.51% | 83.51% | 80.30% | 83.51% | 83.51% | 83.51% | 83.51% | 81.17% |
| | S4L Attack (Jagielski et al., 2020) | 85.99% | 82.04% | 80.72% | 81.86% | 83.72% | 85.49% | 71.23% | 82.80% | 10.00% |
| | Smoothing Attack (Lukas et al., 2022) | 87.86% | 85.27% | 84.07% | 84.81% | 86.16% | 87.37% | 76.26% | 85.36% | 10.00% |
| | D-DAE (Chen et al., 2023b) | 87.32% | 85.62% | 84.82% | 77.30% | 84.81% | 86.96% | 63.79% | 84.97% | 78.21% |
| | D-DAE+ (Chen et al., 2023b) | 87.32% | 86.58% | 86.84% | 84.17% | 84.26% | 86.93% | 58.23% | 84.27% | 50.24% |
| | pBayes Attack (Tang et al., 2024) | 87.32% | 86.58% | 87.20% | 87.13% | 84.04% | 86.70% | 85.63% | 84.63% | 84.24% |
| JBDA-TR | Naive Attack (Truong et al., 2021) | 75.50% | 66.54% | 53.32% | 59.56% | 62.55% | 73.38% | 38.13% | 61.55% | 10.00% |
| | Top-1 (Sanyal et al., 2022) Attack | 62.55% | 62.55% | 62.55% | 53.71% | 62.55% | 62.55% | 62.55% | 62.55% | 61.45% |
| | D-DAE (Chen et al., 2023b) | 75.50% | 55.89% | 40.63% | 55.59% | 59.61% | 67.83% | 16.00% | 61.14% | 47.10% |
| | D-DAE+ (Chen et al., 2023b) | 75.50% | 72.48% | 67.45% | 65.59% | 63.28% | 72.67% | 31.86% | 64.65% | 40.88% |
| | pBayes Attack (Tang et al., 2024) | 75.50% | 70.90% | 67.25% | 74.99% | 63.27% | 74.46% | 63.46% | 66.57% | 65.33% |

Table 12: **Stage 1 (D→D)** Evaluation results on the MedMNIST dataset. We report PSNR, SSIM, MS-SSIM and watermark detection success rate.

| Watermark | PSNR↑ | SSIM↑ | MS-SSIM↓ | WM Succ.↑ |
|---|---|---|---|---|
| DwtDct | 39.51 | 0.972 | 0.993 | 0.7815 |
| DwtDctSvd | 39.44 | 0.974 | 0.993 | 1.0 |
| RivaGAN | 41.22 | 0.963 | 0.992 | 0.6675 |

adopted for evaluating medical foundation models. We design our evaluation around the proposed closed-loop taxonomy, where attacks propagate through data–model interactions. Specifically, we first introduce a watermark removal attack, which removes the watermark through reconstruction regeneration attack (D→D), we then introduce a data poisoning attack (D→M) that corrupts the training distribution and compromises the learned model. Then, we perform model extraction (M→M) to show how to steal a black-box pre-trained model. Finally, we demonstrate how model inversion (M→D) leaks sensitive training data information. We present the details and results in the following.

**Stage 1: Watermark removal attack** In this setting, the data owner embeds imperceptible watermarks into the training data that can only be detected or verified by authorized parties. To evaluate watermark removal attacks, we adopt the threat model of Zhao et al. (2024a), where an adversary attempts to eliminate these hidden signals through data regeneration. Specifically, the attacker reconstructs or regenerates the protected samples in a way that preserves semantic content while erasing the embedded watermark, thereby undermining ownership verification without access to the original watermarking mechanism.

We adopt two representative watermarking schemes: DwtDctSvd (Navas et al., 2008) and RivaGAN (Zhang et al., 2019). To evaluate watermark removal attacks, we apply a set of commonly used distortion and regeneration–based techniques, including JPEG compression, Bmshj2018 (Ballé et al., 2018), and Cheng2020 (Cheng et al., 2020).

Following (Zhao et al., 2024a), we measure the *Peak Signal-to-Noise Ratio (PSNR)*, *Structural Similarity Index (SSIM)*, *Multi-Scale Structural Similarity Index (MS-SSIM)* and watermark detection success rate. Table 12 reports the watermark detection success rates before any attack. Table 13 presents the detection results after applying watermark removal attacks. As shown, watermark removal substantially degrades detection performance, leading to a significant reduction in detection rates.

**Stage 2: Data poisoning attack** We adopt the threat model introduced by (Shafahi et al., 2018), in which the adversary does not require access to or manipulation of training labels. Instead, the attacker subtly injects carefully crafted samples into the training set with the goal of inducing a targeted misbehavior at inference time. Crucially, this manipulation is designed to affect the model's prediction on a specific test instance while leaving its overall predictive performance largely unchanged, thereby remaining difficult to detect. We follow the data poisoning optimization in (Shafahi et al., 2018) and crafted 50 poisoned training data, the performance is shown in Table 14.

Table 13: **Stage 1 (D→D)** Watermark removal performance under different attacks. We report bit accuracy (Bit Acc.) and watermark success rate (WM Succ.).

| Method | Attack | Bit Acc. | WM Succ. |
|---|---|---|---|
| DwtDct | cheng2020 | 0.4825 | 0.0000 |
| | bmshj2018 | 0.4820 | 0.0000 |
| | jpeg | 0.5222 | 0.006 |
| DwtDctSvd | cheng2020 | 0.4871 | 0.0005 |
| | bmshj2018 | 0.4924 | 0.0000 |
| | jpeg | 0.6803 | 0.2475 |
| RivaGAN | cheng2020 | 0.5710 | 0.0015 |
| | bmshj2018 | 0.5864 | 0.003 |
| | jpeg | 0.6795 | 0.2105 |

Table 14: **Stage 2 (D→M) data poisoning impact on the model performance.**

| **Test accuracy** (Original training data) | **Test accuracy** (Poisoned training data) |
|---|---|
| 93.64% | 85.72% |

**Stage 3: Model extraction attack** The objective of a model extraction attack is to replicate the functionality of a deployed black-box model that is accessible only through query-based interactions (e.g., via an API). Following the experimental protocol in (Tang et al., 2024), we assume the attacker has no access to the model's internal parameters or training data, and instead learns a surrogate model solely by querying the target model and observing its outputs. We evaluate this threat by conducting a series of controlled extraction experiments designed to measure how effectively the attacker can approximate the behavior of the original model under limited query access. Table 15 summarizes the effectiveness of model extraction attacks under both soft-label and hard-label query settings. Despite the attacker having access only to black-box query outputs and using a query dataset (TinyImageNet200 and TissueMNIST) that is substantially different from the data used to train the victim model (PathMNIST in MedMNIST dataset (Yang et al., 2021; 2023a)), the extracted student models are still able to achieve non-trivial accuracy, reaching 37.06% and 40.15% under soft-label and hard-label supervision, respectively. These results indicate that, even under significant data distribution mismatch and restricted query access, model extraction attacks can still recover meaningful functional behavior of the target model.

Table 15: **Stage 3 (M→M) model extraction on the victim model.**

| Category | Victim model test accuracy (Teacher) | Clone model test accuracy (Student) |
|---|---|---|
| Soft label | 85.72% | 34.69% |
| Hard label | 85.72% | 36.32% |

**Stage 4: Model inversion attack** The goal of a model inversion attack is to recover sensitive information about the training data by exploiting access to a trained model. Following prior work (Fang et al., 2022), we consider a threat model in which the attacker seeks to reconstruct representative input samples that are highly correlated with the model's predictions. The attack leverages the model's learned decision boundaries and confidence signals to infer properties of the underlying data distribution, thereby revealing private information that was implicitly encoded during training. We evaluate this threat by assessing the accuracy by retraining the model on the reconstructed samples. Table 16 shows the student model accuracy after model inversion.

Table 16: **Stage 4 (M→D) model inversion on the victim model.**

| Teacher model test accuracy | Student model test accuracy |
|---|---|
| 36.32% | 20.18% |

## 10 Conclusion

In summary, this work establishes a unified perspective on machine learning security by framing the interplay between data and models through a closed-loop threat taxonomy. By organizing interactions along four fundamental directions—data-to-data, data-to-model, model-to-data, and model-to-model—our framework reveals how vulnerabilities and defenses are interconnected across the entire ML pipeline. This holistic view not only clarifies existing threat relationships but also provides a foundation for designing more generalizable and resilient defense strategies in the era of large-scale and foundation models.

**Broader Impact**   This survey covers a range of security threats arising from data–model interactions in modern machine learning systems and these attacks have important societal, ethical, and economic implications.

First, membership inference, model inversion, and training data extraction attacks can expose sensitive information about individuals whose data were used during training. Such leakage may violate privacy expectations and consent, and in real-world settings could enable downstream harms such as identity disclosure, discrimination, or misuse of personal data. We emphasize that these risks extend beyond technical methods and directly affect data subjects, highlighting the need for privacy-preserving training practices, responsible data governance, and regulatory compliance alongside algorithmic defenses.

Second, model extraction attacks pose broader economic and innovation risks. By enabling adversaries to replicate proprietary models, these attacks threaten the sustainability of Machine Learning as a Service (MLaaS) business models and may weaken incentives for organizations to invest in developing high quality and socially beneficial AI systems. As extraction techniques become more accessible, there is a growing need for robust defenses, legal frameworks, and deployment time safeguards that balance openness and intellectual property protection.

In summary, this survey emphasizes that AI security is not solely a technical concern. It has direct consequences for individual privacy and the long-term health of the AI ecosystem. Addressing these challenges requires coordinated advances in technical defenses, policy, and responsible deployment practices.

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
