# OpenReview forum: "AI Security in the Foundation Model Era: A Comprehensive Survey from a Unified Perspective"
_TMLR — Accepted by TMLR_

### Review · Reviewer_E7UU · 2025-11-20

**Summary Of Contributions:**

This paper presents a survey on AI security in the era of foundation models, proposing a unified closed-loop threat taxonomy that aims to conceptually and theoretically consolidate various AI security issues. The authors categorize the origins of AI security threats into Data (D) and Models (M), arguing that security challenges in foundation models stem from interactions between these two elements. A detailed classification is introduced, organizing AI security problems into four types (D→D, D→M, M→M, M→D), each accompanied by explanations and formal representations. The survey systematically examines mathematical formulations, technical approaches, and defense strategies for various attacks in multimodal settings, while also highlighting current limitations such as the static nature of defenses. Future directions, including adaptive defense and cross-modal security, are suggested.

Overall, the paper is well-structured, substantive, and offers valuable insights, though minor issues in formatting and formula selection, as well as occasional oversights, warrant slight revisions.

Strengths:

The novel four-type taxonomy of AI security threats provides a clear and unifying framework that could facilitate future research and conceptual descriptions in the field.

The paper offers clear explanations of most AI security issues, making it a helpful resource for newcomers to the field. Necessary formalizations are included to aid understanding of complex problems.

The content is comprehensive, covering a wide range of complex issues related to foundation models—such as their large parameter size, strong generalization capacity, multimodal nature, and tunability—with appropriate breadth and depth.

The analysis breaks down attack and defense cases across multiple modalities rather than focusing solely on text or graph structures. By linking security characteristics of foundation models to emerging threats in fine-tuning and API-based scenarios, the paper addresses practical concerns in real-world AI applications.

Weaknesses:

Text and Formula Issues: Paragraph lengths vary significantly, and formulas are used frequently. Some shorter sections (e.g., 2.2.1 and 2.2.2) could potentially be merged, and formula representations simplified or unified where appropriate.

Coverage Gaps: Certain AI security issues, such as cascading attacks (e.g., model pollution followed by model extraction and data leakage), may not fit neatly into a single category and deserve more attention.

Limited Perspective: The analysis primarily focuses on the "data–model interaction" dimension. Incorporating additional dimensions—such as attack timing (training/inference/deployment) or threat objectives (privacy/integrity/availability)—could enrich the discussion. Differentiating attacks of the same direction but varying risk levels (e.g., untargeted data poisoning vs. targeted backdoor poisoning) would also be beneficial.

Lack of Quantitative Comparisons: The paper would benefit from quantitative experimental results (e.g., attack success rates, defense overhead) to compare different defense methods, rather than relying mainly on qualitative summaries.

Defense Strategy Selection: In cases involving cross-component attacks, it remains unclear how to prioritize defense strategies—whether based on computational cost, effectiveness, real-time performance, or other criteria.

Clarity and Formatting: Certain specialized terms (e.g., inverse model attacks and extraction attacks) could benefit from more accessible explanations. Additionally, some formula references are incomplete, and overall formatting should be reviewed for consistency.

**Audience:**

Yes

**Audience Explanation:**

This paper focuses on "AI Security in the Foundation Model", closely related to the focus of this journal.

**Broader Impact Concerns:**

This paper performs well in this aspect.

**Claims And Evidence:**

Yes

**Claims Explanation:**

I think this paper should be accepted after revision, since it meets the criteria of this paper.

**Requested Changes:**

Text and Formula Issues: Paragraph lengths vary significantly, and formulas are used frequently. Some shorter sections (e.g., 2.2.1 and 2.2.2) could potentially be merged, and formula representations simplified or unified where appropriate.

Coverage Gaps: Certain AI security issues, such as cascading attacks (e.g., model pollution followed by model extraction and data leakage), may not fit neatly into a single category and deserve more attention.

Limited Perspective: The analysis primarily focuses on the "data–model interaction" dimension. Incorporating additional dimensions—such as attack timing (training/inference/deployment) or threat objectives (privacy/integrity/availability)—could enrich the discussion. Differentiating attacks of the same direction but varying risk levels (e.g., untargeted data poisoning vs. targeted backdoor poisoning) would also be beneficial.

Lack of Quantitative Comparisons: The paper would benefit from quantitative experimental results (e.g., attack success rates, defense overhead) to compare different defense methods, rather than relying mainly on qualitative summaries.

Defense Strategy Selection: In cases involving cross-component attacks, it remains unclear how to prioritize defense strategies—whether based on computational cost, effectiveness, real-time performance, or other criteria.

Clarity and Formatting: Certain specialized terms (e.g., inverse model attacks and extraction attacks) could benefit from more accessible explanations. Additionally, some formula references are incomplete, and overall formatting should be reviewed for consistency.

---

> ### Author Response · Authors · 2026-01-03
> **Author Response (1/2)**
>
> We sincerely appreciate the reviewer’s constructive feedback.
>
> **Q1** Text and Formula Issues: Paragraph lengths vary significantly, and formulas are used frequently. Some shorter sections (e.g., 2.2.1 and 2.2.2) could potentially be merged, and formula representations simplified or unified where appropriate.
>
> **A1** Thank you for pointing this out. In the revised version, we have merged several sections to improve clarity and coherence. We also simplified the formulas and consolidated related equations to enhance readability.
>
> **Q2** Coverage Gaps: Certain AI security issues, such as cascading attacks (e.g., model pollution followed by model extraction and data leakage), may not fit neatly into a single category and deserve more attention.
>
> **A2** We appreciate the reviewer’s point. Importantly, cascading attacks such as model pollution → model extraction → data leakage can be naturally decomposed into a sequence of individual attack steps, each of which fits cleanly into one of our four security categories. These categories serve as fundamental building blocks for analyzing more complex, multi-stage threats. These compound attacks illustrate how real-world adversaries may chain together multiple categories, which our framework fully accommodates by allowing sequential attacks.
>
>
> **Q3**  Limited Perspective: The analysis primarily focuses on the "data–model interaction" dimension. Incorporating additional dimensions—such as attack timing (training/inference/deployment) or threat objectives (privacy/integrity/availability)—could enrich the discussion. Differentiating attacks of the same direction but varying risk levels (e.g., untargeted data poisoning vs. targeted backdoor poisoning) would also be beneficial.
>
> **A3**  We thank the reviewer for this insightful suggestion. In the revised manuscript, we explicitly incorporate attack timing as an orthogonal dimension that complements our primary taxonomy based on data–model interaction.
>
>  We add a new subsection “Attack Timing: Training-Time, Inference-Time, and Deployment-Time Attacks” (**Section 7.2**), in which we distinguish training-time, inference-time, and deployment-time attacks. We further introduce **Table 5**, which maps each attack category to the phase(s) at which it occurs.
>
>
> We add a new subsection “Attack Objectives: Privacy, Integrity, and Availability” (**Section 7.1**). While our taxonomy organizes attacks based on their interaction with the data and model components, the classical security objectives—privacy, integrity, and availability—provide an orthogonal lens centered on attacker goals in **Table 4**.
>
> We also added more detailed discussions on untargeted data poisoning vs. targeted backdoor poisoning in **Section 3.1** (Special Case I).
>
>
> This extension makes explicit that our taxonomy is not restricted to data–model interaction, but can be layered with additional dimensions such as timing and attacker objectives. We believe this multi-dimensional view strengthens the survey’s practical value for both researchers and practitioners.
>
> **Q4** Lack of Quantitative Comparisons: The paper would benefit from quantitative experimental results (e.g., attack success rates, defense overhead) to compare different defense methods, rather than relying mainly on qualitative summaries.
>
> **A4** Thank you for your suggestions. We provide quantitative comparisons
> in **Section 9.1 from Table 6-12**, covering the four attack categories.

---

> ### Author Response · Authors · 2026-01-03
> **Author Response (2/2)**
>
> **Q5** Defense Strategy Selection: In cases involving cross-component attacks, it remains unclear how to prioritize defense strategies—whether based on computational cost, effectiveness, real-time performance, or other criteria.
>
>
> **A5**: Thank you for raising this important point. We agree that cross-component attacks naturally complicate defense selection because they span multiple vulnerability surfaces. Our framework is designed to identify and decompose such attacks into their constituent components, which enables practitioners to systematically determine where defenses must be applied.
>
> While our paper does not prescribe a single universal prioritization rule since real-world systems differ in latency budgets, deployment constraints, and threat tolerance.
>
> * Attack decomposition: By decomposing a cross-component attack into its constituent stages and mapping each stage to the corresponding security axis, we can identify the most critical points of failure. This decomposition enables defenders to prioritize and select defense strategies based on the most severe security consequences.
>
> * Defense prioritization by constraints: Depending on system requirements, defense strategies can be prioritized according to constraints such as computational cost, latency, and real-time feasibility. The selection of defenses should reflect the most critical operational requirements.
>
> * Composed mitigation: Multi-stage attacks typically require composed defenses.
>
>
>
> **Q6** Clarity and Formatting: Certain specialized terms (e.g., inverse model attacks and extraction attacks) could benefit from more accessible explanations. Additionally, some formula references are incomplete, and overall formatting should be reviewed for consistency.
>
>
> **A6** Thank you for your suggestions! We have clarified the descriptions of model inversion, training data extraction attacks and model extraction attacks to make them more accessible. Specifically, model inversion attacks are explained in more detail in **Section 4.1** (Special Case I), while training data extraction attacks are discussed in Special Case III. We also provide a clearer illustration of model extraction attacks in **Section 5.1**. In addition, we have revised the formula references and improved the overall formatting for clarity and readability.

---

> > ### Author Response · Authors · 2026-01-12
> > **Response**
> >
> > Dear Reviewer E7UU,
> >
> > We deeply thank you for your insightful comments. We have addressed each concern in the revised manuscript and believe these changes have significantly strengthened the paper. We hope these revisions meet your expectations and adequately address the points raised. Could you please let us know if these changes are satisfactory or if further clarification is needed?  Thank you very much for your time and efforts.

---

### Review · Reviewer_bkdN · 2025-12-19

**Summary Of Contributions:**

1. Summary of Contributions:

This paper presents a survey of AI security threats and defenses for foundation models. The authors propose a unified closed-loop threat taxonomy that categorizes attacks based on the interaction between Data and Model. Taxnomy consist of following 4 classes
1) Data-> Data: Attacks involving direct manipulation of data content such as watermark removal, decryption, jailbreaking
2) Data -> Model: Attacks that corrupt the learning process such as poisoning, harmful fine-tuning.
3) Model->Data: Attacks that extract information from the model about its training data such as membership inference, inversion.
4) Model->Model: Attacks aimed at replicating or stealing model such as model extraction.

The paper also attempts to formalize these interactions mathematically and provides survey of defense agaisnt this threat.


Strengths:
1) Relevance: The survey includes lot of recent literature, covering emerging topics related to FM security
2) Taxonomic Organization: The 4 class taxonomy provides a high-level heuristic for categorizing the diverse landscape of FM security.

Weakness:
1) The classification of Jailbreak Attacks under D->D is logically flawed. Jailbreaking is fundamentally an attack on a model’s alignment, not just a data transformation. yet the authors group it with data decryption and watermark removal, which are fundamentally different operations.
2) The authors frequently cite arXiv preprints versions of papers that have already been published in major conferences/Journals.
3) A core claim of the paper is offering a Unified closed-loop perspective that exposes interdependencies and shared principles. However, the paper fails to substantively discuss these connections. The body of the text treats threats in isolation where each section covers different classes such as D->D, D->M etc, with almost no analysis of how an attack in one class specifically facilitates an attack in another. The promise of a unified view is unfulfilled.
4) There are several typos and formatting inconsistencies

**Audience:**

Yes

**Audience Explanation:**

The survey aggregates a large number of recent papers relevant to the security risks in LLMs and provided taxonomy. This paper will help TMLR audience a quick summary of the field to get caught up quickly.

**Claims And Evidence:**

No

**Claims Explanation:**

As discussed in the weakness:

A core claim of the paper is offering a Unified closed-loop perspective that exposes interdependencies and shared principles. However, the paper fails to substantively discuss these connections. This paper treats threats in isolation where each section covers different classes such as D->D, D->M etc, with almost no analysis of how an attack in one class specifically facilitates an attack in another. The promise of a unified view is unfulfilled.

**Requested Changes:**

Summary of changes:

1) To justify the unified closed-loop title, the authors must add a dedicated section or substantial discussion analyzing the connections between the classes. For example, explain how Poisoning attacks can degrade privacy defenses against attacks. Without this, the closed-loop claim is unsupported.

2) The authors must update citations to their peer-reviewed versions if it's exist rather than citing arXiv preprints. and check if all the paper exist
For example:
a. Yunhao Chen, Shujie Wang, Difan Zou, and Xingjun Ma. Extracting training data from unconditional diffusion models. arXiv preprint arXiv:2410.02467, 2024. -> This paper has been withdrawn by Yunhao Chen from arxiv
b. Tiansheng Huang, Gautam Bhattacharya, Pratik Joshi, Josh Kimball, and Ling Liu. Antidote: Postfine-tuning safety alignment for large language models against harmful fine-tuning. arXiv preprint
arXiv:2408.09600, 2024a -> Tiansheng Huang, Gautam Bhattacharya, Pratik Joshi, Joshua Kimball, Ling Liu Proceedings of the 42nd International Conference on Machine Learning, PMLR 267:25059-25074, 2025.
c. Tiansheng Huang, Sihao Hu, Fatih Ilhan, Selim Furkan Tekin, and Ling Liu. Booster: Tackling harmful finetuning for large language models via attenuating harmful perturbation. arXiv preprint arXiv:2409.01586,
2024c -> https://openreview.net/forum?id=tTPHgb0EtV

3) The authors must move Jailbreak Attacks to a category that reflects the interaction with the model likely Data->Model. If it remains in Data->Data, the justification must be rigorously rewritten.

4) Typos, grammar and Formatting Consistency. Please revisit this throughout paper:
a) Repeated names in citations, such as "Dong et al. Dong et al." (Page 6) , "Li et al. Li et al." (Page 10)
b) The goal of watermark removal attacksZhao et al. (2024a) -> No space
c) Fix the grammatical error in the introduction: "As The growth..." should be "The growth..." or "As the growth...."

---

> ### Author Response · Authors · 2026-01-03
> **Author Response (1/2)**
>
> We deeply thank the reviewer for the valuable comments and suggestions.
>
> **Q1** The classification of Jailbreak Attacks under D->D is logically flawed. Jailbreaking is fundamentally an attack on a model’s alignment, not just a data transformation. yet the authors group it with data decryption and watermark removal, which are fundamentally different operations.
>
>
> **A1** We thank the reviewer for raising this important point. We moved jailbreak attacks to the data->model (D->M) category as **special case III of D->M attack in Section 3**. We also reformulated jailbreak attack in the revision.
>
>
>
> **Q2** The authors frequently cite arXiv preprints versions of papers that have already been published in major conferences/Journals.
>
>
> **A2** Thank you for pointing out this! We have corrected the arxiv papers citations and cited their conference/journal versions in the revision.

---

> ### Author Response · Authors · 2026-01-03
> **Author Response (2/2)**
>
> **Q3**  A core claim of the paper is offering a Unified closed-loop perspective that exposes interdependencies and shared principles. However, the paper fails to substantively discuss these connections. The body of the text treats threats in isolation where each section covers different classes such as D->D, D->M etc, with almost no analysis of how an attack in one class specifically facilitates an attack in another. The promise of a unified view is unfulfilled.
>
>
> **A3** Thank you for your questions. We discussed attacks interaction in **Section 6.2**, Attack Interactions and Inter-dependencies in the revision. We also discussed attacks interaction in the following
>
>
> A key insight of our unified closed-loop perspective is that security threats in foundation models do not operate in isolation. Instead, attacks across the data space (D) and model space (M) are interdependent, forming feedback loops in which vulnerabilities in one class systematically facilitate, amplify or weaken threats in another. We identify several cross-class connections that substantiate this closed-loop unified view.
>
>
> **Poisoning (D$\rightarrow$M) $\rightarrow$ Membership Inference (M$\rightarrow$D)** Data poisoning attacks manipulate the training or fine-tuning data to corrupt the learned feature--label relationships encoded in model parameters (D$\rightarrow$M). Beyond degrading predictive performance, such corruption can amplify privacy attacks. In particular, poisoned representations can increase overfitting, making it easier for adversaries to exploit output statistics for membership inference (M$\rightarrow$D) [1,2]. The resulting privacy leakage further illustrates how model manipulation in the training phase can manifest as data exposure at inference time.
>
> **Model Extraction (M$\rightarrow$M) $\rightarrow$ Membership Inversion, Training Data Recovery (M$\rightarrow$D) and Adversarial Data Generation (D$\rightarrow$D)** Successful model extraction yields a high-fidelity surrogate model that approximates the behavior of the target system (M$\rightarrow$M). This surrogate not only enables training data recovery attacks and model inversion (M$\rightarrow$D) [3], but can also be used to synthesize adversarial data samples that closely follow the original training data distribution (D$\rightarrow$D) [4]. Such adversarially generated data may subsequently bypass data filtering or safety checks, illustrating the interaction between model compromise and data-space exploitation.
>
>
> **Model Inversion (M$\rightarrow$D) $\rightarrow$ Watermark Removal (D$\rightarrow$D)** Model inversion attacks aim to reconstruct training data or data-like samples from a deployed model. Although the recovered samples may not be exact replicas of the original training instances, they often preserve salient semantic and distributional characteristics of the underlying data.   As a result, an adversary can retrain a new model using inversion-generated data and claim that the model was trained solely on independently obtained, *legitimate* data since it is not directly traceable to the original training set. This effectively enables watermark removal and ownership obfuscation, as the resulting model may evade existing IP verification or data provenance mechanisms [5].
>
>
>
>
> **Backdoor Attacks (D$\rightarrow$M) weakens Model Extraction (M$\rightarrow$M)** [6] proposes a defensive strategy that intentionally poisons the model extraction process by embedding a backdoor into any surrogate model trained from stolen outputs. Instead of preventing queries, the defender modifies the victim model’s output so that its predictions subtly encode malicious signals while preserving normal task accuracy for benign users. When an attacker performs model extraction using these outputs, the resulting surrogate unintentionally learns a hidden backdoor, which can later be triggered to induce targeted misbehavior. This effectively makes model extraction vulnerable to controlled misuse and demonstrates that backdoor mechanisms can serve as an active defense against model extraction by poisoning the downstream learning process.
>
>
> Reference:
>
> [1] Amplifying membership exposure via data poisoning, NeurIPS 2022
>
> [2] Privacy backdoors: Enhancing membership inference through poisoning pre-trained models, NeurIPS 2024
>
> [3] tealing machine learning models via prediction {APIs}, USENIX Security 16, 2016
>
> [4] Practical black-box attacks against machine learning, Asia CCS, 2017
>
> [5] Ipremover:
> A generative model inversion attack against deep neural network fingerprinting and watermarking, AAAI 2024.
>
> [6] Honeypotnet: Backdoor attacks against model extraction, AAAI 2025
>
>
> **Q4**  There are several typos and formatting inconsistencies
>
>
>
> **A4** Thank you for pointing out this. We have corrected typos and formatting inconsistencies in the revision.

---

> ### Author Response · Authors · 2026-01-12
> **Response**
>
> Dear Reviewer  bkdN,
>
> We deeply thank you for your insightful comments. We have addressed each concern in the revised manuscript and believe these changes have significantly strengthened the paper. We hope these revisions meet your expectations and adequately address the points raised. Could you please let us know if these changes are satisfactory or if further clarification is needed? Thank you very much for your time and efforts.

---

### Review · Reviewer_8WWD · 2025-12-22

**Summary Of Contributions:**

This survey paper proposes a unified closed-loop threat taxonomy for AI security that organizes the landscape of attacks and defenses along four directional axes based on data-model interactions: Data→Data (D→D), Data→Model (D→M), Model→Data (M→D), and Model→Model (M→M). The central statement is that data and models in machine learning systems are deeply interdependent, and security threats should not be studied in isolation but rather understood as interconnected phenomena that can propagate vulnerabilities across the ML pipeline.

The authors accomplish three main things with this work. First, they aim to provide a comprehensive survey that brings together the fragmented literature on ML security by viewing attacks through the lens of data-model interactions—essentially arguing that we should stop treating poisoning, extraction, and inference attacks as separate research silos. Second, they systematically walk through the major attack families and their defenses, trying to surface the common threads and recurring patterns that cut across different threat types. Third, they step back to identify where the field still has gaps and where future research efforts might be most productively directed, particularly as we move deeper into the foundation model era.

Strengths of the paper:
1, Novel Organizational Framework. The closed-loop taxonomy viewing security through bidirectional data-model interactions is a fresh perspective that helps conceptualize how different attacks relate to each other. This is more principled than existing surveys that typically organize attacks by attack type alone without considering their interdependencies.

2, Comprehensive Literature Coverage. The survey covers an impressive breadth of literature (200+ references) spanning multiple modalities (vision, language, audio, graphs) and attack families. The coverage extends from classical adversarial attacks to recent foundation model vulnerabilities, making it a valuable reference document.

3, Consistent Mathematical Treatment. The authors provide unified mathematical formulations for each attack category, establishing formal connections between different attack types through a general optimization framework. This formalization helps clarify the underlying objectives and constraints of various attacks.

4, Timely Focus on Foundation Models. The emphasis on foundation model security is highly relevant given current deployment trends. The paper appropriately highlights how model scale, fine-tuning APIs, and multimodal capabilities introduce new attack surfaces that traditional security frameworks do not adequately address.

Weaknesses of the paper:
1, The grouping of jailbreak attacks under D→D is conceptually problematic. Jailbreaks fundamentally involve model interaction—the crafted input's purpose is to manipulate model behavior, not to transform data independently. This appears more aligned with D→M dynamics, and the justification provided ("protection-bypass attacks") feels forced and undermines the framework's coherence.

2, The paper asserts that threats are "deeply interconnected" and form "dynamic chains of influence," but provides no empirical evidence, case studies, or quantitative analysis demonstrating these claimed interdependencies. Without validation, the framework remains a conceptual contribution without proven practical utility.

3, The paper lacks summary tables comparing attack characteristics, threat models, or defense effectiveness. Such comparisons would significantly enhance the survey's utility as a reference and help readers quickly identify relevant work for their specific concerns.

**Audience:**

No

**Audience Explanation:**

1, The TMLR's community should care about this paper because security is no longer a niche concern—it is becoming central to how we deploy models in practice. As foundation models proliferate across healthcare, finance, legal systems, and consumer applications, understanding what can go wrong is not optional. Practitioners who fine-tune GPT-4 through an API or deploy open-weight models need to grasp the attack surface they are inheriting. This paper attempts to map that terrain comprehensively, which has genuine practical value even if the execution is imperfect.

2, One legitimate motivation behind this work is that ML security research has become balkanized. Researchers working on backdoor attacks often do not engage deeply with the membership inference literature, and those studying model extraction may have limited awareness of watermarking defenses. This fragmentation means insights do not transfer across subfields as readily as they should. By placing these different attack families within a common framework, the paper offers a shared vocabulary and conceptual scaffolding that could facilitate cross-pollination. Whether or not readers fully buy the closed-loop framing, having everything laid out side-by-side helps researchers identify connections they might otherwise miss.

3, The paper's attention to harmful fine-tuning attacks is particularly timely. The dominant paradigm now involves taking pre-trained foundation models and adapting them through fine-tuning APIs or parameter-efficient methods like LoRA. This creates attack vectors that did not exist when models were trained from scratch in controlled environments. The ML community needs to understand that safety alignment can be undone with surprisingly few examples, and that even benign-looking fine-tuning data can degrade model behavior. This paper consolidates recent findings on these vulnerabilities in a way that should raise awareness.

4, Even if the claimed interdependencies are not rigorously demonstrated, the underlying intuition matters for how we think about defenses. If poisoning a model really does make it more vulnerable to inversion attacks, then defending against one threat in isolation may provide false confidence. The paper pushes readers to think about defense in depth rather than point solutions. This systems-level perspective—viewing data and model security as entangled rather than separable—is intellectually valuable and could influence how researchers approach defense design going forward.

**Broader Impact Concerns:**

I have two concerns:
1, The M→D section covers membership inference, model inversion, and training data extraction in substantial detail. These attacks can expose sensitive personal information, violate consent expectations of individuals whose data was used in training, and potentially enable downstream harms like identity theft or discrimination. The paper treats these as technical problems without discussing the human impact on data subjects whose information might be extracted. A broader impact discussion should acknowledge that the attacks surveyed can cause real harm to real people.

2, Model extraction attacks threaten the economic viability of ML-as-a-service businesses and could undermine incentives for developing beneficial AI systems. While the paper covers defenses, it does not discuss the broader economic and innovation ecosystem implications of these threats becoming more widely understood and executed.

**Claims And Evidence:**

No

**Claims Explanation:**

While the paper provides solid evidence for its descriptive claims about individual attack and defense mechanisms, the more ambitious claims about the framework's unifying power and the interconnected nature of threats remain largely unsupported assertions.

Claims That Are Well-Supported:
1, Coverage of individual attack families. The paper does an adequate job surveying the existing literature within each category. The descriptions of poisoning attacks, model extraction, membership inference, and other established threats are accurate and appropriately cite relevant work. Readers seeking an overview of what attacks exist and how they operate will find reasonable coverage backed by appropriate references.

2, Mathematical formulations. The optimization-based framing of different attacks is technically sound. The general objective in Equation (1) and its specializations to different attack scenarios are mathematically coherent and accurately reflect how these problems are formulated in the source literature.

Claims That Lack Convincing Evidence:
1, The "fragmented literature" problem. The authors assert that existing surveys treat threats in isolation, but they never systematically demonstrate this gap. A convincing argument would compare what prior surveys cover, show specific blind spots, and explain how this framework addresses them. Instead, the claim is simply stated in the introduction without substantiation.

2, Interdependencies between attack categories. This is perhaps the paper's central claim—that threats "form a dynamic chain of influence wherein the compromise of one component can recursively propagate vulnerabilities throughout the ML pipeline." The authors offer illustrative examples (e.g., "a poisoning attack can corrupt the feature-label relationship, thereby weakening the model's robustness to inversion"), but these are hypothetical statements without empirical backing. No experiments, case studies, or citations to work demonstrating such attack chains are provided. This undermines the core value proposition of the unified framework.

3, Framework utility. The paper claims the taxonomy "offers a principled lens for analyzing and defending foundation models" and "establishes a foundation for developing scalable, transferable, and cross-modal security strategies." However, no evidence demonstrates that viewing attacks through this lens actually leads to better defenses or deeper insights than existing organizational schemes. The framework is presented but never validated.

4, Missing comparative evaluation. The paper does not compare its framework against alternative organizational schemes to show that the D→D, D→M, M→D, M→M taxonomy captures something that other frameworks miss.

5, No case studies. A single detailed case study tracing how a real-world attack propagated across the proposed categories would substantially strengthen the paper's central statement. Its absence is notable.

**Requested Changes:**

Suggestions for adjustments:
1, Reconsider or Rigorously Justify the D→D Categorization. The inclusion of jailbreak attacks under D→D is the most significant conceptual problem with the framework. Jailbreaks fundamentally operate by manipulating model behavior through crafted inputs—this is squarely in the territory of data affecting models. The authors must either:
a)	Move jailbreak attacks to a more appropriate category (perhaps D→M or a new hybrid category)
b)	Provide a substantially more convincing argument for why jailbreaks constitute data-to-data transformations rather than data-to-model influence
c)	Acknowledge this as a limitation and explain the tradeoffs involved in the current placement
This is required for acceptance.

2, Provide Empirical or Case-Based Evidence for Claimed Interdependencies. The paper's central value proposition rests on the claim that threats are interconnected and form dynamic chains of influence. This claim is currently unsupported. The authors should:
a)	Include at least two or three detailed case studies demonstrating how attacks in one category enable or amplify attacks in another
b)	Cite existing empirical work that documents such attack chains, if available
c)	Alternatively, conduct small-scale experiments showing, for example, that a poisoned model leaks more information under inversion attacks than a clean model
This is required for acceptance.

3, Add Comparison Tables. A survey of this scope needs summary tables to be useful as a reference. At minimum, the authors should include:
a)	A table comparing attack categories along dimensions such as threat model assumptions, attacker capabilities required, typical success metrics, and computational costs
b)	A table mapping defenses to the attacks they address, with notes on effectiveness and limitations
c)	A comparison with prior surveys showing what this work covers that others do not
This is required to Strengthen the paper.

4, Add a Running Example. Threading a concrete running example through all four categories—perhaps involving a hypothetical healthcare ML system—would help readers understand how the framework applies in practice and how threats might chain together.
This is required to Strengthen the paper.

5, Strengthen the Future Directions Section. The current discussion identifies broad areas but lacks specificity. The authors should:
a)	Articulate concrete open problems that researchers could immediately pursue
b)	Identify specific technical barriers that need to be overcome
c)	Suggest evaluation methodologies or benchmark datasets that would advance the field
This is required to Strengthen the paper.

---

> ### Author Response · Authors · 2026-01-03
> **Author Response**
>
> We are grateful to the reviewer for the insightful comments and suggestions.
>
> **Q1** Reconsider or Rigorously Justify the D→D Categorization. The inclusion of jailbreak attacks under D→D is the most significant conceptual problem with the framework. Jailbreaks fundamentally operate by manipulating model behavior through crafted inputs—this is squarely in the territory of data affecting models. The authors must either: a) Move jailbreak attacks to a more appropriate category (perhaps D→M or a new hybrid category) b) Provide a substantially more convincing argument for why jailbreaks constitute data-to-data transformations rather than data-to-model influence c) Acknowledge this as a limitation and explain the tradeoffs involved in the current placement This is required for acceptance.
>
> **A1** We thank the reviewer for pointing out this important issue. In response, we have reclassified jailbreak attacks under the data-to-model (D→M) category, treating them as a **special case III** within this attack class in **Section 3**. We also  reformulated the jailbreak attack formulation in the revision.

---

> ### Author Response · Authors · 2026-01-03
> **Author Response**
>
> **Q2** Provide Empirical or Case-Based Evidence for Claimed Interdependencies. The paper's central value proposition rests on the claim that threats are interconnected and form dynamic chains of influence. This claim is currently unsupported. The authors should: a) Include at least two or three detailed case studies demonstrating how attacks in one category enable or amplify attacks in another b) Cite existing empirical work that documents such attack chains, if available c) Alternatively, conduct small-scale experiments showing, for example, that a poisoned model leaks more information under inversion attacks than a clean model This is required for acceptance.
>
>
> **A2** Thank you for pointing out this!  We included additional examples and discussed attacks interaction in **Section 6.2**, Attack Interactions and Inter-dependencies in the revision. We also discussed attacks interaction in the following.
>
>
> A key insight of our unified closed-loop perspective is that security threats in foundation models do not operate in isolation. Instead, attacks across the data space (D) and model space (M) are interdependent, forming feedback loops in which vulnerabilities in one class systematically facilitate, amplify or weaken threats in another. We identify several cross-class connections that substantiate this closed-loop unified view.
>
>
> **Poisoning (D$\rightarrow$M) $\rightarrow$ Membership Inference (M$\rightarrow$D)** Data poisoning attacks manipulate the training or fine-tuning data to corrupt the learned feature--label relationships encoded in model parameters (D$\rightarrow$M). Beyond degrading predictive performance, such corruption can amplify privacy attacks. In particular, poisoned representations can increase overfitting, making it easier for adversaries to exploit output statistics for membership inference (M$\rightarrow$D) [1,2]. The resulting privacy leakage further illustrates how model manipulation in the training phase can manifest as data exposure at inference time.
>
> **Model Extraction (M$\rightarrow$M) $\rightarrow$ Membership Inversion, Training Data Recovery (M$\rightarrow$D) and Adversarial Data Generation (D$\rightarrow$D)** Successful model extraction yields a high-fidelity surrogate model that approximates the behavior of the target system (M$\rightarrow$M). This surrogate not only enables training data recovery attacks and model inversion (M$\rightarrow$D) [3], but can also be used to synthesize adversarial data samples that closely follow the original training data distribution (D$\rightarrow$D) [4]. Such adversarially generated data may subsequently bypass data filtering or safety checks, illustrating the interaction between model compromise and data-space exploitation.
>
>
> **Model Inversion (M$\rightarrow$D) $\rightarrow$ Watermark Removal (D$\rightarrow$D)** Model inversion attacks aim to reconstruct training data or data-like samples from a deployed model. Although the recovered samples may not be exact replicas of the original training instances, they often preserve salient semantic and distributional characteristics of the underlying data.   As a result, an adversary can retrain a new model using inversion-generated data and claim that the model was trained solely on independently obtained, *legitimate* data since it is not directly traceable to the original training set. This effectively enables watermark removal and ownership obfuscation, as the resulting model may evade existing IP verification or data provenance mechanisms [5].
>
>
>
>
> **Backdoor Attacks (D$\rightarrow$M) weakens Model Extraction (M$\rightarrow$M)** [6] proposes a defensive strategy that intentionally poisons the model extraction process by embedding a backdoor into any surrogate model trained from stolen outputs. Instead of preventing queries, the defender modifies the victim model’s output so that its predictions subtly encode malicious signals while preserving normal task accuracy for benign users. When an attacker performs model extraction using these outputs, the resulting surrogate unintentionally learns a hidden backdoor, which can later be triggered to induce targeted misbehavior. This effectively makes model extraction vulnerable to controlled misuse and demonstrates that backdoor mechanisms can serve as an active defense against model extraction by poisoning the downstream learning process.
>
>
> Reference:
>
> [1] Amplifying membership exposure via data poisoning, NeurIPS 2022
>
> [2] Privacy backdoors: Enhancing membership inference through poisoning pre-trained models, NeurIPS 2024
>
> [3] tealing machine learning models via prediction {APIs}, USENIX Security 16, 2016
>
> [4] Practical black-box attacks against machine learning, Asia CCS, 2017
>
> [5] Ipremover: A generative model inversion attack against deep neural network fingerprinting and watermarking, AAAI 2024.
>
> [6] Honeypotnet: Backdoor attacks against model extraction, AAAI 2025

---

> ### Author Response · Authors · 2026-01-03
> **Author Response**
>
> **Q3** Add Comparison Tables. A survey of this scope needs summary tables to be useful as a reference. At minimum, the authors should include: a) A table comparing attack categories along dimensions such as threat model assumptions, attacker capabilities required, typical success metrics, and computational costs b) A table mapping defenses to the attacks they address, with notes on effectiveness and limitations c) A comparison with prior surveys showing what this work covers that others do not This is required to Strengthen the paper.
>
>
> **A3**  Thank you for your constructive suggestions. In the revised manuscript, we have added three new tables to strengthen clarity:
>
> * (a) We present a comprehensive comparison of major attack categories in terms of their threat model assumptions, required attacker capabilities, evaluation metrics, and computational cost (**Table 2**) in the revision.
>
> * (b) We provide a systematic mapping between defense mechanisms and the attack types they are designed to mitigate, including their effectiveness and limitations (**Table 3**) in the revision.
>
> * (c) We include a comparative analysis between our survey and prior surveys, highlighting the unique coverage and perspectives offered by our work that are not addressed in existing literature (**Table 1**) in the revision.
>
>
>
> **Q4** Add a Running Example. Threading a concrete running example through all four categories—perhaps involving a hypothetical healthcare ML system—would help readers understand how the framework applies in practice and how threats might chain together. This is required to Strengthen the paper.
>
> **A4** Thank you for your suggestions. To illustrate AI security in practice, we construct a medical image classification pipeline using the PathMNIST in MedMNIST dataset, a large-scale and standardized benchmark widely adopted for evaluating medical foundation models. We design our evaluation around the proposed closed-loop taxonomy, where attacks propagate through data–model interactions. Specifically, we first introduce a watermark removal attack, which removes the watermark through reconstruction regeneration attack (D$\rightarrow$D), we then introduce a data poisoning attack (D$\rightarrow$M) that corrupts the training distribution and compromises the learned model. Then, we perform model extraction (M$\rightarrow$M) to show how to steal a black-box pre-trained model. Finally, we demonstrate how model inversion (M$\rightarrow$D) leaks sensitive training data information. More details and results are presented in **Section 9.2** in the revision.

---

> ### Author Response · Authors · 2026-01-03
> **Author Response**
>
> **Q5** Strengthen the Future Directions Section. The current discussion identifies broad areas but lacks specificity. The authors should: a) Articulate concrete open problems that researchers could immediately pursue b) Identify specific technical barriers that need to be overcome c) Suggest evaluation methodologies or benchmark datasets that would advance the field This is required to Strengthen the paper.
>
> **A5**  Thank you for your suggestions! We have revised the future directions in **Section 8**.  We also discussed the future works in the following:
>
> This survey advocates a *closed-loop* view of AI security in the era of foundation models, where data- and model-centric attacks are deeply interconnected rather than isolated. While existing studies have made important progress on individual threats, real-world deployments increasingly face *simultaneous, adaptive, and composed attacks*. Below, we outline concrete and open research directions that move beyond single-attack analysis and toward holistic, system-level defenses.
>
> ---
>
> ## **Defending Against Simultaneous and Composed Attacks**
>
> **Open Problems** Most existing defenses are designed and evaluated against a *single attack* (e.g., extraction *or* jailbreak *or* inversion). A critical open problem is how to design defenses that are *robust to multiple attacks occurring concurrently or sequentially*. Several important questions remain unresolved:
>
> * How can a defense mitigate multiple attack objectives simultaneously without assuming prior knowledge of which attack is active?
> * How should defenses adapt when attackers jointly optimize multiple goals, such as maintaining jailbreak success while extracting model information?
>
> **Technical Barriers** Defending against multiple attacks raises several fundamental challenges:
>
> * **Multiple defense interference:** Defenses optimized for one threat can degrade robustness against others (e.g., adversarial training may inadvertently amplify the effectiveness of model extraction attacks [1]). This arises because many attacks share internal representations, so mitigating one vulnerability can unintentionally weaken protection against another.
> * **Conflicting objectives:** Security, alignment, privacy, and utility objectives are often incompatible, making joint optimization difficult.
>
> **Evaluation Methodology** We argue that future work should adopt *multi-attack testing* protocols, where:
>
> * Multiple attacks are launched concurrently or sequentially (e.g., jailbreak + model extraction, data poisoning → membership inference).
> * Defenses are evaluated using joint metrics such as worst-case attack success, cross-attack amplification, and utility degradation.
> * At least one defense-aware adaptive multi-attack adversary is included to assess robustness under strategic adaptation.
>
>
> ## **Closed-Loop Defense Design and Adaptation**
>
> **Open Problems** A promising direction is to move from static defenses to *closed-loop adaptive defenses* that evolve with model updates and attacker behavior. Key open problems include:
>
> 1. Online detection of attacker behavior drift under non-stationary query distributions.
> 2. Learning defense policies that dynamically choose interventions (e.g., refusal, perturbation, detection) under utility and latency constraints.
> 3. Continual recalibration of defenses after fine-tuning, alignment updates, or retrieval-augmented deployment.
>
> **Technical Barriers** Key challenges include delayed feedback about attack success, high false-positive rates under distribution shift, and the risk that adaptive defenses introduce new vulnerabilities.
>
> **Evaluation Methodology** We recommend streaming evaluation protocols that measure detection success rate, cumulative harm, and long-term utility under adaptive attackers rather than static test sets.
>
>
>
> ## **Joint Optimization of Security, Alignment, and Utility**
>
> **Open Problems** Most defenses implicitly optimize a single axis (e.g., privacy or alignment). A central open challenge is how to *jointly optimize* security, alignment, and utility. Future work should formalize multi-objective optimization frameworks that explicitly expose trade-offs and develop harm-aware metrics that go beyond attack success rate to capture downstream impact.
>
> **Technical Barriers** Major obstacles include the lack of composable guarantees across different defense mechanisms and the mismatch between existing metrics and real-world harm.
>
> **Evaluation Methodology** Future evaluations should report full trade-off curves, including utility loss, compute overhead, robustness under adaptive attacks, failure severity, and alignment degradation.
>
>
>
> ## **Benchmarking**
>
> A major gap is the absence of unified benchmarks that reflect closed-loop and multi-attack realities. We envision comprehensive benchmarks that explicitly model attack chaining across data and model stages, along with standardized reporting of attacker budgets, access levels, and defense costs.

---

> ### Author Response · Authors · 2026-01-03
> **Author Response**
>
> ## Broader Impact Concerns:
>
> **Q** I have two concerns: 1, The M→D section covers membership inference, model inversion, and training data extraction in substantial detail. These attacks can expose sensitive personal information, violate consent expectations of individuals whose data was used in training, and potentially enable downstream harms like identity theft or discrimination. The paper treats these as technical problems without discussing the human impact on data subjects whose information might be extracted. A broader impact discussion should acknowledge that the attacks surveyed can cause real harm to real people.
>
> 2, Model extraction attacks threaten the economic viability of ML-as-a-service businesses and could undermine incentives for developing beneficial AI systems. While the paper covers defenses, it does not discuss the broader economic and innovation ecosystem implications of these threats becoming more widely understood and executed.
>
> **A** We added broader impact in **Section 10** in the revision, the details are presented in the following.
>
> This survey covers a range of security threats arising from data–model interactions in modern machine learning systems and these attacks have important societal, ethical, and economic implications.
>
> First,  membership inference, model inversion, and training data extraction attacks can expose sensitive information about individuals whose data were used during training. Such leakage may violate privacy expectations and consent, and in real-world settings could enable downstream harms such as identity disclosure, discrimination, or misuse of personal data. We emphasize that these risks extend beyond technical methods and directly affect data subjects, highlighting the need for privacy-preserving training practices, responsible data governance, and regulatory compliance alongside algorithmic defenses.
>
> Second, model extraction attacks pose broader economic and innovation risks. By enabling adversaries to replicate proprietary models, these attacks threaten the sustainability of Machine Learning as a Service (MLaaS) business models and may weaken incentives for organizations to invest in developing high quality and socially beneficial AI systems. As extraction techniques become more accessible, there is a growing need for robust defenses, legal frameworks, and deployment time safeguards that balance openness and intellectual property protection.
>
> In summary, this survey emphasizes that AI security is not solely a technical concern. It has direct consequences for individual privacy and the long-term health of the AI ecosystem. Addressing these challenges requires coordinated advances in technical defenses, policy, and responsible deployment practices.

---

> ### Author Response · Authors · 2026-01-12
> **Response.**
>
> Dear Reviewer 8WWD,
>
> We deeply thank you for your insightful comments. We have addressed each concern in the revised manuscript and believe these changes have significantly strengthened the paper. We hope these revisions meet your expectations and adequately address the points raised. Could you please let us know if these changes are satisfactory or if further clarification is needed?  Thank you very much for your time and efforts.

---

### Decision · Action_Editor_yHYz · 2026-01-25

**Recommendation:** Accept as is

**Audience:**

Yes

**Audience Explanation:**

This work is highly relevant to the TMLR audience, as the security of foundation models is a critical and rapidly evolving area of concern for both researchers and practitioners.

**Claims And Evidence:**

Yes

**Claims Explanation:**

The submission provides a theoretically grounded and comprehensive survey of the AI security domain for foundation models.